**Investigation**

# Analysis of lifespan across diversity outbred mouse studies identifies multiple longevity-associated loci

Martin N. Mullis,[1] Kevin M. Wright,[1] Anil Raj,[1] Daniel M. Gatti (ID) ,[2] Peter C. Reifsnyder,[2] Kevin Flurkey,[2] Jonathan R. Archer,[2] Laura Robinson,[2] Andrea Di Francesco,[1] Karen L. Svenson,[2] Ron Korstanje (ID) ,[2] David E. Harrison,[2] J. Graham Ruby,[1] Gary A. Churchill (ID) [2,*]

[1]Calico Life Sciences LLC, 1170 Veterans Blvd, South San Francisco, CA 94080, United States
[2]The Jackson Laboratory, 600 Main St, Bar Harbor, ME 04609, United States

*Corresponding author: The Jackson Laboratory, 600 Main St, Bar Harbor, ME 04609, United States. Email: gary.churchill@jax.org

Lifespan is an integrative phenotype whose genetic architecture is likely to highlight multiple processes with high impact on health and aging. Here, we conducted a genetic mega-analysis of longevity in Diversity Outbred (DO) mice that included 2,444 animals from 3 independently conducted lifespan studies. We identified 8 loci that contributed significantly to lifespan independently of diet and drug treatment in at least one study. One of these loci also influenced lifespan in a sex-dependent manner, and we detected an additional locus with a diet-specific effect on lifespan. Collectively, these loci explained over half of the estimated heritable variation in lifespan across these studies and provided insight into the genetic architecture of lifespan in DO mice.

Keywords: genetics; lifespan; aging; diversity outbred mice; quantitative trait analysis; QTL mapping

## Introduction

Lifespan is a quantitative phenotype that can summarize health and fitness information across populations and physiological systems. When seeking to discover factors (e.g. phenotypes, environment effects, or genetic loci) that are potentially causative, the integrative nature of this phenotype provides both advantages and disadvantages to researchers. Disadvantages relate to the plethora of independent contributors: each correlated factor likely makes only a small contribution to lifespan, limiting the statistical power to discover it. When discovered, the non-specific nature of lifespan also confounds interpretation: the correlated factor could affect any unknown sub-population or subset of health phenomena. On the other hand, those correlates that do emerge are likely to have either the highest and/or broadest impacts on overall health for the population, which provides an advantage in terms of prioritization. These principles apply to any covariate of lifespan, including genetic variants with effects on longevity.

In human populations, the heritability ($h^2$) of lifespan is low, consistently reported to be under 20% (Van Den Berg *et al.* 2017; Kaplanis *et al.* 2018) and under 10% when accounting for assortative mating (Ruby *et al.* 2018). This limits the power of quantitative trait locus (QTL) discovery. Large cohorts have nonetheless empowered such discovery, but the significance of many loci vary across distinct populations and study designs (Fortney *et al.* 2015; McDaid *et al.* 2017; Wright *et al.* 2019). Outbred mouse populations provide orthogonal resources to study the genetics of mammalian lifespan, with advantages and disadvantages vs human populations. The limited sample sizes of typical mouse experiments (ranging from dozens of animals to hundreds) vs

modern human cohorts are a disadvantage (Logan *et al.* 2013; Gatti *et al.* 2014; Smallwood *et al.* 2014; French *et al.* 2015). Advantages include the following: well-balanced allele frequencies, which increase statistical power to detect loci; lack of substantial population structure, which confounds genotype/phenotype correlations in human populations but has minimal impacts in mouse populations (Churchill *et al.* 2012; Ghazalpour *et al.* 2012; Wang *et al.* 2021); and the ability to enroll mice at a young age and then follow them until the end of their natural lifespans, facilitating prospective analyses of lifespan. Finally, treatments and environmental perturbations may be applied to mouse populations, allowing the measurement of their effects on lifespan and other health-related phenotypes.

Numerous lifespan-associated QTL have been mapped in yeast (Stumpferl *et al.* 2012), nematodes (Ayyadevara *et al.* 2003; Snoek *et al.* 2019), flies (Highfill *et al.* 2016), and rodents using a variety of techniques (Hook *et al.* 2018). These include the use of congenic mouse strains to investigate the potential effects of particular loci (Smith and Walford 1977), with subsequent research utilizing backcrosses (Yunis *et al.* 1984). Recombinant inbred lines (Peirce *et al.* 2004) and intercrosses (Yuan *et al.* 2013) have also been used to identify lifespan QTL, but those study populations contained limited genetic diversity. Advances in genome sequencing and the development of large-scale intercrosses (Miller *et al.* 2007; Bou Sleiman *et al.* 2022) and outbred populations (Churchill *et al.* 2012; Svenson *et al.* 2012) are now beginning to enable the mapping of lifespan in populations with greater genetic diversity.

Several interventions are known to extend mouse lifespan (Miller *et al.* 2007) and healthspan, including multiple approaches to dietary restriction (Weindruch *et al.* 1986; Anderson *et al.* 2009;

Mitchell *et al.* 2019). Caloric restriction is the most widely-studied form of dietary restriction and typically refers to a 20%–40% reduction in calorie intake. This form of dietary restriction has been shown to extend lifespan in organisms ranging from yeast to rodents, although the effects of caloric restriction vary by species and the time at which the intervention is started (Speakman and Mitchell 2011; Taormina and Mirisola 2014). Several compounds have also been shown to extend lifespan in multiple species, including the drug rapamycin, which was originally studied as an immunosuppressant and anti-cancer drug but later found to modulate cellular growth and metabolism through its effects on mTOR (mechanistic Target of Rapamycin; Heitman *et al.* 1991; Harrison *et al.* 2009; Bjedov *et al.* 2010; Johnson *et al.* 2013). Caloric restriction and rapamycin both impact metabolic/nutrient-sensing pathways and are robust across mouse genotypes, as evidenced by their validation in genetically heterogeneous populations (Harrison *et al.* 2009; Di Francesco *et al.* 2024). Despite their robustness, consensus on the physiological mechanisms-of-action for these interventions is lacking (Papadopoli *et al.* 2019; Green *et al.* 2022). Mechanistic insight could be gained through study of interacting effects between genetic loci and either of these interventions, considered as environment covariates (gene-by-environment interactions; GxE; Ottman 1996). Genetic loci responsive to dietary restriction have been mapped in mice for traits related to cardiac physiology (Zhang *et al.* 2022) as well as lifespan in flies (Pallares *et al.* 2023). GxE effects involving diet and lifespan have also been reported in BXD strains (Roy *et al.* 2021; Williams *et al.* 2022). However, GxE effects involving caloric restriction or rapamycin treatment remain unexplored at the locus level in diverse, outbred populations.

Here, we present data from 3 previously unpublished lifespan studies conducted using Diversity Outbred (DO) mice (Churchill *et al.* 2012): the Harrison, Svenson, and Shock studies. We used these data, together with DO lifespan data from the published Dietary restriction (DRiDO) study (Di Francesco *et al.* 2024), to evaluate the robustness of lifespan-extending interventions and characterize the genetic architecture of mouse lifespan. We found dietary restriction and rapamycin treatment to robustly increase lifespan across all relevant studies (Harrison, Svenson and DRiDO). For the 3 studies that included genotypes (Harrison, Shock and DRiDO), we performed genetic analysis of each individual study, identifying 5 non-overlapping QTL associated with lifespan. Further mega-analysis across all 3 genetic studies revealed 3 additional QTL. Finally, we performed a genome-wide GxE scan, revealing one locus whose effect on lifespan was modified by dietary intervention.

## Methods
### Study designs
#### Dietary restriction (DRiDO)

This study is extensively described elsewhere (Di Francesco *et al.* 2024). Briefly, female DO mice were received at ~4 weeks of age in 12 waves from March 2016 through November 2017. Mice were housed in groups of 8 in single large-format ventilated pens with nestlets, biotubes, and gnawing blocks. Mice were fed a standard chow diet (5K0G, LabDiet). Surviving mice were randomized to 1 of 5 dietary interventions, which were initiated for the surviving mice at 6 months of age: ad libitum (AL; $n = 188$), 1 day/week fasting (1D; $n = 188$), 2 days/week fasting (2D; $n = 190$), 20% caloric restriction (20; 2.75 g/mouse/day of chow; $n = 189$), and 40% caloric restriction (40; 2.06 g/mouse/day of chow; $n = 182$).

Mice were extensively phenotyped as described (Di Francesco *et al.* 2024) and maintained until they died naturally. The mouse room was on a 12/12 h light/dark schedule from 6:00 AM to 6:00 PM and kept at $73° \pm 2°$ F.

#### Harrison

Founder DO mice (167 retired breeder pairs) were obtained from the Jackson Laboratory and female offspring were accumulated for the lifespan study over 5 months. All mice were microchipped at 4 weeks of age. Mice were housed 22 per large-format double pens connected by a tunnel on an open-air rack. All 22 mice per pen were from different breeder pairs. Pens had pine shaving bedding with acidified water. Every week, one pen of the connected pair would be changed. Mice would be herded into one pen and the tunnel blocked off. The dirty pen would be removed, and a new clean pen attached with fresh water and grain. The ad libitum control mice ($n = 349$) were on a non-irradiated diet (5LG6, or "5S84", TestDiet, Purina) from weaning. The diet restricted ($n = 335$) mice received 2.2 g/day/mouse of ground non-irradiated diet via modified fish feeders that were programmed to dump the ground diet onto the floor of the cage between 6 and 7 PM after lights were off. Modified feeders were restocked every 7 days. Any grain left in the feeders after 7 days was dumped on the cage floor. Proper feeder performance was indicated by a weighted string that was wound around a screw when the feeders dumped food. Diet restriction began at 4 weeks of age after being microchipped. An additional 339 mice received non-irradiated diet until they were 16 months of age, whereupon they started on 5LG6 diet with 142 ppm encapsulated rapamycin (Rapamycin Holdings, actual concentration of rapamycin in diet is 14 ppm, TestDiet, Purina). Mice were maintained until they died naturally. The mouse room was on a 12/12 h light/dark schedule from 6:00 AM to 6:00 PM and kept at $73° \pm 2°$ F.

#### Svenson

We obtained female DO mice from the Jackson Laboratory breeding colony at ~4 weeks of age. Mice were obtained in 8 waves over the course of 1 year and enrolled by randomization to dietary intervention protocols. Mice were housed 8 per group in single large format pens. Interventions were implemented as described for the Harrison study with a few differences indicated here. The ad libitum fed control mice ($n = 319$) were on a 4% irradiated diet (5K52, aka "5KOG", TestDiet, Purina). The calorie restricted mice ($n = 316$) received 2.2 g/day/mouse of ground 4% irradiated diet. CR mice were fed at ~7 AM daily and food was placed directly onto the bottom of the pen by a technician. On Friday, the CR mice received a triple feeding (6.6 g/mouse) and were fed again on Monday morning. A third group of mice ($n = 317$) were maintained on the ad libitum protocol until 16 months of age and were then switched to rapamycin diet as described above. Mice experienced minimal handling (monthly body weights and weekly pen changes) and were maintained until they died naturally. The mouse room was on a 12/12 h light/dark schedule from 6:00 AM to 6:00 PM and kept at $70° \pm 2°$ F.

For this study, DNA samples were collected for genotyping on the MUGA array, as described for other studies below. However, irregularities with sample labeling/handling made us question the integrity of our ID matches between mice and samples. We performed a quality-control assessment by comparing genotype-predicted vs recorded coat colors across the mice (Silvers 2012) and confirmed extensive sample mismatches (data not shown). We therefore excluded these data from our genetic analyses.

### Shock

We obtained female ($n = 244$) and male ($n = 240$) DO mice from the Jackson Laboratory breeding colony at ∼4 weeks of age. Mice were obtained in 5 waves from June 2011 through August 2012. Mice were housed in single-sex groups of 5 in standard ventilated duplex pens. All mice were fed ad libitum on 6% sterilized grain (5K52, aka "5KOG", TestDiet, Purina). Mice experienced minimal handling (body weights and other non-invasive procedures). At 6, 12, and 18 months, we obtained $3 \times 100$ µL retroorbital blood draws, with 2 weeks recovery time between each. Mice were maintained until they died naturally. The mouse room was on a 12/12 h light/dark schedule from 6:00 AM to 6:00 PM and kept at $70° \pm 2°$ F. We performed a quality-control assessment of genotyping using animal sex, similar to the method described above, and confirmed that genotyping data matched our records.

All procedures used in these studies were reviewed and approved by the Jackson Laboratory Animal Care and Use Committee.

## Genotyping

Genotypes for all studies reported here were obtained using the mouse universal genotyping array (MUGA; Morgan *et al.* 2016). DNA was isolated from tail tips using standard methods and shipped to Neogen Genomics (Lincoln, NE, USA) for analysis. Samples were genotyped using the MUGA (Harrison), MegaMUGA (Shock), or GigaMUGA (DRiDO) genotyping arrays. Founder haplotypes were reconstructed using the R/qtl2 software, and samples with call rates at or above 90% were retained for analysis. Genome coordinates were from mouse genome GCRm39, and gene locations were taken from the Mouse Genome Informatics databases (Blake *et al.* 2021). To facilitate mega-analysis across datasets sequenced using different MUGA arrays, genotype probabilities were first computed by applying the Hidden Markov model in R/qtl2 to platform-specific marker loci, followed by a linear interpolation of the genotype probabilities to a set of 69,005 pseudo-markers. Finally, the interpolated genotype probabilities were converted to haplotype probabilities via R/qtl2.

## Data analysis

### Survival analysis

We compared survival among each cohort of animals, as well as between experimental groups within studies. This was done by plotting Kaplan–Meier curves and by testing the equivalence of survival distributions among each cohort or experimental group using log-rank tests using overall tests (across cohorts and within each cohort) as well as pairwise comparisons between each experimental group and its respective within-study control group (ex: comparing rapamycin treatment to ad libitum within the Harrison study). In each study, analysis begins at the time of intervention. *P*-values are reported with no correction for multiple comparisons and are considered significant at $P < 0.05$. Median lifespan was estimated in each cohort as well as within each experimental group. The effects of dietary interventions and/or sex were estimated via Cox proportional hazards regression analysis and are reported as hazard ratios with 95% confidence intervals. *P*-values are reported without correction for multiple comparisons and are considered significant at $P < 0.05$. Survival analysis was conducted using the "survival" (Therneau and Grambsch 2000; Therneau 2024) package in R and plotted via the "ggsurvfit" package (Sjoberg *et al.* 2024). Mortality doubling times and baseline hazards were estimated, beginning at the time of intervention, from a Gompertz log-linear hazard model with a 95% confidence interval and percentage change relative to female mice on an ad libitum diet via the "flexsurv" package in R (Jackson 2016).

### Additive whole-genome scans

All genetic analysis was conducted using the "qtl2" package in R (Broman *et al.* 2019; R Core Team 2024). Whole-genome scans for lifespan QTL were carried out via the scan1() function using a mixed effects model in which lifespan was regressed on 8-state allele probabilities for each individual in a dataset and LOD scores were recorded at each marker (Broman and Sen 2009). Within the Dietary Restriction and Harrison studies, dietary intervention and DO generation were included as additive covariates. In the Shock study, Sex and DO generation were included as additive covariates. In the mega-analysis, Study, Diet, Sex, and DO generation were included as additive covariates. In each whole-genome scan, kinship was included in the linear mixed effects model as a random effect. For each genome-wide scan, 1,000 permutations of the data were performed in which phenotypes were randomized and a whole-genome scan was run (Kassambara and Mundt 2020). The maximum LOD score observed in each permuted scan was recorded, and the 95th percentile of the distribution of 1,000 maximum LOD scores was used as the significance threshold ($\alpha = 0.05$). QTL with LOD scores greater than this threshold were considered significant at our permutation-based threshold. In addition to this significance level, an false discovery rate (FDR)-based significance level of LOD ≥ 6 was also used to identify loci contributing to variance in lifespan. We report 2LOD support intervals ('2LOD SI'), corresponding to a 2LOD drop around each sentinel marker identified in whole-genome scans.

### FDR analysis

To assess the FDR of significance thresholds used in GWAS and mega-analysis, we performed 1,000 permutations of each dataset in which phenotypes were randomized and whole-genome scans were run. Across a range of LOD scores, QTLs were identified using the *find_peaks()* function in the "qtl2" package in each permutation of the data (false discoveries) and in the non-permuted data (actual discoveries). In each permutation, the number of false discoveries at a particular LOD score was divided by the total number of discoveries (false discoveries + actual discoveries) to calculate the FDR, and the mean FDR across the 1,000 permutations was recorded. This process was repeated for each LOD score to generate the plot shown in Supplementary Fig. 1.

While less conservative than the permutation-based thresholds, the secondary threshold of LOD ≥ 6 corresponds to a mean FDR of 0.169 in individual studies (DRiDO $FDR_{LOD6} = 0.129$, Harrison $FDR_{LOD6} = 0.165$, Shock $FDR_{LOD6} = 0.213$) and an FDR of 0.091 in the mega-analysis (Supplementary Fig. 1). Adjusting LOD thresholds to correspond to an FDR of 0.15 in individual studies does not impact the results presented in the analysis, while adjusting the FDR to 0.1 across individual studies results in a failure to detect 2 QTLs on chromosome 7 in the DRiDO study.

### Forward regression analysis

In the mega-analysis, forward regression analysis was performed to account for the effects of genome-wide significant QTL when searching for additional loci influencing lifespan. This was done via the scan1() function using a mixed effects model including dietary intervention, sex, and DO generation as additive effects and kinship as a random effect. In addition to these covariates, previous QTL identified at a genome-wide significance level were included in the model as additive effects. QTLs were encoded as numeric variables representing the genotype state at the marker

with the highest LOD score as reported by association mapping. Only QTL reaching permutation-based significance thresholds were included in the model as additive covariates.

### Effect size estimation and percent variance explained

Best linear unbiased predictors (BLUPs) and corresponding standard errors were computed for all QTL using the scan1blups() function in "rqtl2" using the additive covariates listed above. Phenotypic variance explained by each QTL was calculated using the following formula (Broman and Sen 2009):

$$1 - 10^{-(2/n)*LOD}$$

where $n$ is the number of samples in a particular dataset and LOD corresponds to the LOD score of the peak marker at each QTL.

### Variant association/fine mapping

Variant association mapping was conducted within a 2LOD support interval of the peak position associated with each QTL. Fine mapping was performed via the scan1snps() function in "rqtl2" using the same additive covariates listed in the *Additive whole-genome scans* section above. In the association mapping process, genotype probabilities for each SNP are estimated based on their strain distribution patterns (SDP) and the mean allele probabilities at the 2 adjacent markers. Variants within 1 LOD of the sentinel SNP within each support interval are considered "most likely" and are highlighted in respective association mapping figures. Variant and gene SQLite datasets used in this analysis are available at the "rqtl2" user guide website: https://kbroman.org/qtl2/assets/vignettes/user_guide.html.

These data are also available at the Jackson Laboratory DO reference data website: https://www.jax.org/research-and-faculty/genetic-diversity-initiative/tools-data/diversity-outbred-reference-data.

### Single-QTL models for diet- and sex-specific loci

Within individual studies, QTLs were tested for interaction with experimental factors unique to those studies. In the Shock cohort, QTLs were tested for interaction with sex, while in other cohorts, QTLs were tested for interactions with one or more of the dietary interventions. Interaction tests were not conducted genome-wide using "rqtl2", since this package does not fit interactions as random effects. For each QTL, tests were run using the 8-state allele probabilities at peak positions identified in whole-genome scans using the fit1() function in "rqtl2". To assess significance, 2 models were run: (1) an additive model in which experimental factors (sex, dietary intervention) and DO generation were included as additive covariates along with kinship as a random effect and (2) an interaction model that included the same additive and random covariates with an additional interaction term corresponding to the experimental factor being tested. The reported LOD is the LOD of the interaction model minus the LOD of the additive model. To establish significance, 1,000 permutations were run at each QTL in which phenotypes were randomized before running the additive and interaction models. After ordering the 1,000 resulting LOD scores, the 95th percentile was chosen as the significance threshold ($\alpha = 0.05$). Interactions between QTL and experimental conditions were considered significant if their LOD was greater than the $\alpha = 0.05$ threshold. Interaction effects were plotted as residual lifespan values as a function of sex after correcting for additive covariates.

### Genome-wide scans for diet- and sex-specific loci

Gene by environment mixed effects models (GxEMM) were run using the "do-qtl" software package in Python version 3.8.16 (Wright *et al.* 2022). Study, sex, and diet were included in the model as fixed effects, while diet and kinship were supplied as random effects. $P$ values at individual markers were computed based on permutations of the data. For the additive term, a significance threshold of Benjamini–Hochberg adjusted $P < 0.01$ was used, corresponding to a $-\log_{10}(P) > 4.85$ (Benjamini and Hochberg 1995). For diet and sex interaction terms, a nominal threshold of $-\log_{10}(P) > 4$ was used to define loci and Benjamini–Hochberg adjusted $P$ values are reported.

## Results

Lifespan data analyzed here derived from 4 separate studies of DO mice, referred to herein as the Harrison study, the Svenson study, the Jackson Laboratory Nathan Shock Center (Shock) study, and the Dietary Restriction in Diversity Outbred mice (DRiDO) study. Three are reported here for the first time: Harrison, Svenson, and Shock. The fourth was previously described (Di Francesco *et al.* 2024). These studies were all conducted independently and involved different cohorts, facilities, technicians, and methodologies (Supplementary Data 1). Each study also had its own design, investigating dietary, drug, and/or sex effects on lifespan. Male mice ($n = 240$) were only included in the Shock study, which examined lifespan in both sexes and did not include any interventions. The vast majority of lifespan data collected in these studies comes from female mice ($n = 2,912$) on either an ad libitum diet, one of several dietary interventions, or a rapamycin-supplemented diet. The treatment groups present in each study are summarized in Fig. 1b–e, and the corresponding experimental designs are described in more detail in the Methods. Genotype data was additionally collected from the Harrison, Shock, and DRiDO studies (see Methods).

Despite varied designs, one control group was shared by all 4 studies: female mice fed an ad libitum ('AL'), nutrient-rich chow diet (Methods). Median lifespans for this category were 794 days (95% CI = 754, 855), 832 days (95% CI = 800, 860), 891 days (95% CI = 863, 922), and 765 days (95% CI = 728, 839) in the Shock, Svenson, Harrison, and DRiDO studies, respectively, and may reflect differences in the design and execution of the studies, including housing, feeding time, and facility (Fig. 1a, Methods). Lifespan in AL females varied significantly by study (log-rank test, $\chi^2 = 33.54$, $P = 2.48 \times 10^{-7}$, Fig. 1a); however, not all of the studies were statistically different from one another (Supplementary Table 1), and study explained a significant but small proportion of total variance in lifespan among AL females (ANOVA, PVE = 1.3%, $P = 2.9 \times 10^{-3}$). Relative to the DRiDO study, the Cox hazard ratio was insignificant for the Shock and Svenson studies (0.88 and 0.85, respectively; $P = 0.2$ and 0.078, respectively), but hazard was significantly reduced in the Harrison study (0.61, $P = 2.03 \times 10^{-7}$). For downstream statistical analyses, study was included as a covariate wherever relevant.

### Rapamycin and caloric restriction both extended lifespan in DO mice

The Harrison and Svenson studies enrolled 1,023 and 952 female mice, respectively, and both probed the effects of 30% caloric restriction ('30CR') and rapamycin drug intervention ('Rapa') on lifespan. Rapamycin did not increase median lifespan relative to AL controls, likely because treatment began in midlife at the age of 16 months. However, beyond this point, rapamycin increased

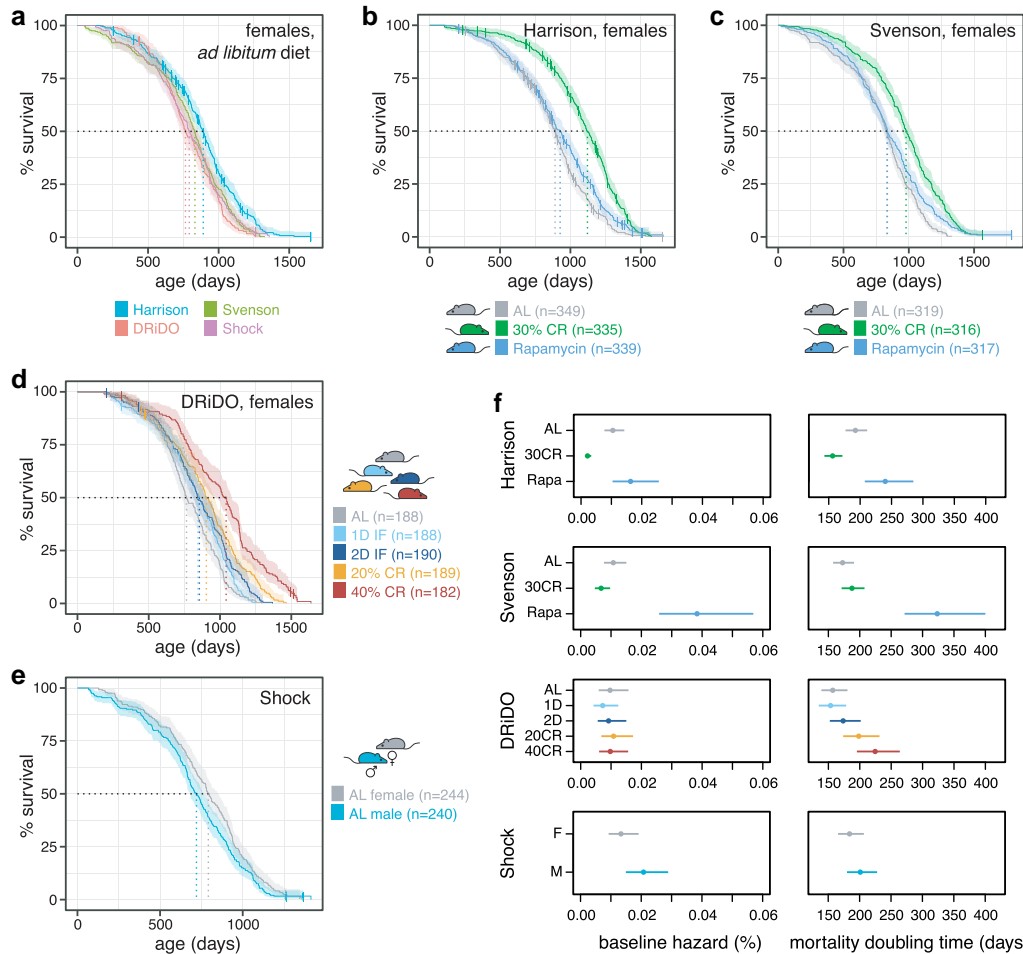

**Fig. 1.** Survival in DO mice across 4 independent studies. a) Kaplan–Meier survival curves for ad libitum females in each of the 4 studies. Dashed vertical lines indicate median survival in days and bands around each curve indicate 95% confidence intervals. Vertical bars on each curve indicate censorship events. b) Kaplan–Meier survival curves by intervention in the Harrison study, which involved female mice on ad libitum (AL), 30% calorie restricted (30% CR), or rapamycin-treated (Rapamycin) diets. c) Kaplan–Meier survival curves by intervention in the Svenson study, which involved female mice on AL, 30% calorie restricted (30% CR), or rapamycin-treated (Rapamycin) diets. d) Kaplan–Meier survival curves by dietary intervention in the DRiDO study Di Francesco et al. (2024), which involved female mice on AL, 1 day intermittent fasting (1D IF), 2 day intermittent fasting (2D IF), 20% calorie restricted (20% CR), or 40% calorie restricted (40% CR) diets. e) Kaplan–Meier survival curves by sex in the Shock study, which included male and female mice on an AL diet. f) Baseline hazard and mortality doubling times and 95% confidence intervals estimated by a Gompertz log-linear model.

lifespan relative to the AL controls in both the Harrison study (Fig. 1b; log-rank test, $\chi^2 = 8.23$, $P = 4.13 \times 10^{-3}$) and the Svenson study (Fig. 1c; log-rank test, $\chi^2 = 11.14$, $P = 8.44 \times 10^{-4}$). Nonetheless, Cox hazard ratios for rapamycin-treated animals vs AL controls were 0.77 (95% CI = 0.66, 0.90) and 0.74 (95% CI = 0.63, 0.86) in the Harrison and Svenson studies, respectively.

Caloric restriction at 30% also increased lifespan in both the Harrison and Svenson studies (Fig. 1b, c). Median lifespan was increased by 230 days in the Harrison study (891 days for AL vs 1,121 days for 30CR; a 25.8% increase) and 150 days (832 for AL vs 982 days for 30CR; an 18% increase) in the Svenson study. This lifespan extension was statistically significant in both studies (log-rank tests, Harrison $\chi^2 = 96.14$, $P = 1.07 \times 10^{-22}$; Svenson $\chi^2 = 75.72$, $P = 3.28 \times 10^{-18}$). Finally, mortality hazard ratios were decreased by similar magnitudes (Harrison Cox HR = 0.47, 95% CI = 0.4–0.56; Svenson Cox HR = 0.52, 95% CI = 0.44, 0.61), suggesting CR had similar effects on longevity despite significant differences in lifespan among AL females in the Harrison and Svenson studies (Supplementary Table 1).

The DRiDO study measured lifespans for 937 female mice randomly assigned to 5 different diet groups at 6 months of age: ad libitum ('AL'), 1 and 2 consecutive days of intermittent fasting ('1D" and "2D", respectively), and 20 and 40% caloric restriction ('20CR" and "40CR", respectively). As previously published (Di Francesco et al. 2024) and depicted in Fig. 1d, all of these variations on dietary restriction significantly increased lifespan.

## Lifespans were longer for female vs male DO mice

The Shock study enrolled 244 female and 240 male mice and did not include any dietary or drug intervention groups: all mice were fed an AL diet similar to the Harrison, Svenson, and DRiDO studies (see Methods). The median lifespan of female mice was 75 days longer than male mice (Fig. 1e; 794 days for females vs 719 days for males; log-rank test, $\chi^2 = 5.33$, $P = 2.09 \times 10^{-2}$), with mortality hazard greater in males (Cox HR = 1.24, 95% CI = 1.03–1.48). This result was reminiscent of the slight female survival advantage reported for genetically diverse UM-HET3 mice, although sex differences in survival are labile to diet, temperature, housing conditions, and strain background (Austad 2011; Masoro and Austad 2011; Cheng et al. 2019; Bou Sleiman et al. 2022).

## The effects of rapamycin and caloric restriction on rates of demographic aging were inconsistent between studies

The mortality doubling time is defined as the rate of increase in mortality hazard with increasing age (Finch *et al.* 1990) and is expressed here as the period of time across which mortality hazard doubles when lifespans are fit to a Gompertzian log-linear hazard model. It was previously reported that CR slowed demographic aging in the DRiDO study for the 20 and 40% CR groups, without having a significant effect on the baseline hazard (Di Francesco *et al.* 2024).

We estimated both the rates of demographic aging and baseline hazard for each sex and intervention group across all 4 studies (Fig. 1f). Although 30% CR extended lifespan reproducibly in both the Harrison and Svenson studies, the effects on the mortality doubling time and baseline hazard were different between studies. In the Harrison study, CR decreased the mortality doubling time by 20% from 192.5 to 156.1 days ($P = 5.83 \times 10^{-4}$) while decreasing baseline hazard from 0.010 to 0.002% ($P = 3.83 \times 10^{-9}$). In the Svenson study, however, there was no significant effect on either the mortality doubling time ($MDT_{AL} = 176.2$ days, $MDT_{CR} = 187.6$ days; $P = 0.16$) or the baseline hazard (hazard$_{AL} = 0.011$%, hazard$_{CR} = 0.007$; $P = 0.11$). These results also contrasted from previously reported effects of 20 and 40% caloric restriction in the DRiDO study, in which caloric restriction significantly reduced the mortality doubling time with no significant effect on baseline hazard (Fig. 1f; Di Francesco *et al.* 2024).

The effects of rapamycin treatment on aging rate and baseline hazard were also inconsistent across study (Fig. 1f). Despite increasing lifespan in both studies, rapamycin treatment did not significantly impact baseline hazard (16 month hazard$_{AL} = 0.014$%, 16 month hazard$_{CR} = 0.016$%; $P = 0.63$) or mortality doubling time (16 month $MDT_{AL} = 209.4$ days, 16 month $MDT_{CR} = 240.0$ days; $P = 0.20$) in the Harrison study. In contrast, in the Svenson study, rapamycin treatment significantly increased the mortality doubling time from 172.6 days to 323.4 days ($P = 1.33 \times 10^{-10}$) while also significantly increasing baseline hazard from 0.011 to 0.038% relative to *ad lib* mice beginning at the time of intervention ($P = 7.49 \times 10^{-7}$). In the Shock study, neither baseline hazard ($P = 0.075$) nor demographic aging rate ($P = 0.31$) significantly varied with sex.

### Study-specific genome-wide association analyses revealed 5 non-overlapping lifespan QTL

To investigate the genetic basis of lifespan, 1009, 460, and 916 mice were genotyped from the Harrison, Shock, and DRiDO studies, respectively (Supplementary Data 2–7; Svenson study mice were not genotyped, see Methods). We used residual maximum likelihood (REML; Broman *et al.* 2019) to estimate the narrow-sense heritability ($h^2$) of lifespan in each study after correcting for sex (Shock) or intervention (Harrison and DRiDO studies), as well as DO generation wave, which may impact allele frequencies and recombination density. Genetics explained 18.5% (SE = 6.5%) of variance in lifespan in the Harrison study and 24.6% (SE = 7.7%) in the DRiDO study. In the Shock study, genetics explained 15.0% (SE = 15.0%) of variance in lifespan, indicating a large degree of uncertainty. Estimates of $h^2$ differed substantially between males and females in this study, with genetics explaining 30.5% of variance in lifespan in females and no significant variance in males ($h^2 = 6.1 \times 10^{-13}$, SE = $8.7 \times 10^{-12}$), suggesting that the survival advantage seen in females in the Shock study may have been driven by technical or experimental factors, such as co-housing.

Estimates of heritability did not significantly differ among diet groups in the Harrison and DRiDO studies (Supplementary Table 2).

Genome-wide scans for quantitative trait loci (QTL) performed separately on lifespans from each of the Harrison, Shock, and DRiDO studies identified 3 unique QTL at a permutation-based significance threshold (Fig. 2a–c; $\alpha = 0.05$, 1,000 permutations; Supplementary Table 3; Doerge and Churchill 1996). Two of these were identified in only one study: chromosome 12 at 82.92 Mb (2LOD support interval, or "2LOD SI" = 81.29–83.37; DRiDO study) and chromosome 16 at 7.11 Mb (2LOD SI = 5.84–8.07; Shock study). Overlapping loci on chromosome 18 were identified from the DRiDO study (21.68 Mb; 2LOD SI = 20.46–25.82) and Harrison study (21.67 Mb; 2LOD SI = 15.81–25.76). At a lower, but still conservative, significance threshold of LOD ≥ 6 (Methods, Supplementary Fig. 1), 2 additional QTLs were identified from the DRiDO study (Fig. 2c): both on chromosome 7, at 11.39 Mb (2LOD SI = 6.76–12.71) and 108.57 Mb (2LOD SI = 101.3–142.98; Supplementary Fig. 2).

For each QTL, the allelic effect of each founder haplotype on lifespan was estimated. The non-overlapping QTL on chromosome 18 had the largest effects on lifespan, explaining 4.7% of phenotypic variance in the DRiDO study and 4.5% of variance in the Harrison study. The effects of these loci were primarily driven by negative effects of the CAST/EiJ (CAST) allele. Animals with one or more CAST alleles at this locus lived 83.4 fewer days on average in the Harrison study (9.0%) and 99 fewer days on average in the DRiDO study (11.8%, Fig. 2d). We presume these to represent the same true QTL, as the effects of each founder haplotype were significantly correlated in each study (Pearson correlation = 0.763, $P = 0.027$, Fig. 2e). In the Shock study, this locus failed to reach genome-wide statistical significance; the effects of the founder haplotypes were significantly correlated with the DRiDO study (Pearson correlation = 0.820, $P = 0.013$) but not the Harrison study (Pearson correlation = 0.557, $P = 0.151$) and the effect of the CAST allele on lifespan at this site was approximately half as large as in the DRiDO or Harrison studies (Fig. 2d, e).

The QTL on chromosomes 12 and 16 explained 4.2 and 6.9% of variance in lifespan within the DRiDO and Shock studies, respectively. Mice containing one or more CAST alleles at the chromosome 12 locus lived 62.1 days longer (7.4%) on average in the DRiDO study (Fig. 2f). In the Shock study individuals with at least one Watkins Star Line B (WSB) allele at the chromosome 16 locus lived 80.1 fewer days (10.8%) on average, and individuals with at least one PWK allele lived 69 days longer on average (9.3%, Fig. 2g).

For each study, we divided the total fraction of phenotypic variance explained by the detected QTL by the variance explained by genetics overall (i.e. the heritability) to infer the mapped percentage of heritable variation. Considering only QTL significant at a genome-wide threshold, 24.3% (0.045/0.185), 46.0% (0.069/0.150), and 36.2% (0.089/0.246) of heritable variation in lifespan was explained in the Harrison, Shock, and DRiDO studies, respectively. Inclusion of the 2 QTL in the DRiDO study on chromosome 7, which each explained 3.2% of the total variance in lifespan in the DRiDO study, increased the percentage of explained heritable variation in this cohort to 62.2% (0.153/0.246). The portion of lifespan variances attributable to genetics but remaining unmapped were 14.0%, 8.1%, and 9.3% in the Harrison, Shock, and DRiDO studies, respectively.

### Mega-analysis of DO lifespan identifies 3 additional QTL

To improve power to detect QTL associated with longevity in DO mice, we performed a mega-analysis of lifespan using combined data from the 2,395 mice in the Harrison, Shock, and DRiDO

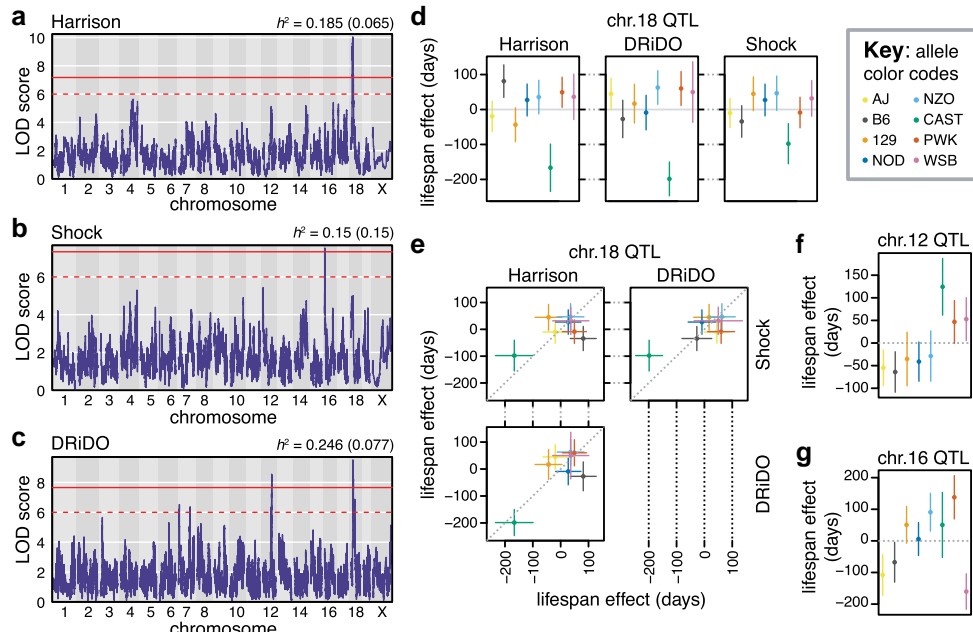

**Fig. 2.** Genetic analysis of lifespan in DO mice. a) QTL profile of the additive genome-wide scan on lifespan in the Harrison study. The solid horizontal line indicates genome-wide significance, while the dashed horizontal line indicates a significance threshold of LOD 6. The narrow-sense heritability ($h^2$) of lifespan is reported, with standard error about the $h^2$ estimate reported in parentheses (*top right*). b) QTL profile of lifespan in the Shock study. c) QTL profile of lifespan in the DRiDO study. d) Allelic effects (BLUPs) and corresponding standard errors of the chromosome 18 locus detected in the Harrison (*left*) and DRiDO (*middle*) studies. The allelic effects at this locus are also shown in the Shock study (*right*), although it did not reach statistical significance in that dataset. e) Comparisons of allelic effects (BLUPs) and standard errors of the chromosome 18 locus in each pair of studies. f) Allelic effects (BLUPs) and corresponding standard errors of the chromosome 12 locus detected in the DRiDO study. g) Allelic effects (BLUPs) and corresponding standard errors of the chromosome 16 locus detected in the DRiDO study.

studies (Supplementary Data 8–9). After correcting for study, sex, intervention, and generation wave, $h^2$ of combined lifespan measurements across the 3 cohorts was 18.6% (SE = 3.8%). A whole-genome scan identified 2 previously identified QTL, chromosome 12 position 81.19 (2LOD SI = 74.9–84.64) and chromosome 18 at 21.81 Mb (2LOD SI = 20.29–25.38) at genome-wide significance (Fig. 3a; Supplementary Table 3). The chromosome 16 locus, previously identified in the Shock cohort, was also detected at a significance level of LOD ≥ 6 at 8.82 Mb (2LOD SI = 6.46–10.46). While neither of the QTLs on chromosome 7 were reproducibly detected in this mega-analysis, novel loci on chromosomes 2 (59.34 Mb; 2LOD SI = 11.04–159.99) and 16 (83.26 Mb; 2LOD SI = 79.84–84.56) were detected at our lower significance threshold. Additionally, a forward scan was performed in which each of the genome-wide significant loci were added to the mixed effects model as additive covariates (Fig. 3b). The forward scan identified an additional locus on chromosome 4 at 137.61 Mb (2LOD SI = 88.47–147.61). The loci on chromosomes 18 and 12 were reproducibly detected using an independent method, GxEMM (Wright *et al.* 2022), at a Benjamini–Hochberg corrected $P$ value threshold of 0.001 ($-\log_{10}(P) > 4.85$; Fig. 3c).

The effects of chromosomes 18 and 12 appeared to be driven by the same alleles detected using data in the DRiDO study, with the CAST allele associated with a decrease in lifespan of 175 days at chromosome 18 and an increase in lifespan of 106 days at the chromosome 12 locus (Fig. 3d, e). Additional QTL displayed more complex patterns of allelic effects, with 2 or more alleles influencing lifespan at these sites (Fig. 3f–i). Mice carrying at least one copy of CAST at the chromosome 18 site had significantly decreased lifespans (log-rank test, $\chi^2 = 40.53$, $P = 1.58 \times 10^{-9}$) and mice carrying at least a single copy of CAST at the chromosome 12 locus had significantly increased lifespans (log-rank test, $\chi^2 = 22.36$,

$P = 1.39 \times 10^{-05}$). After correcting for study, diet, sex, and DO generation, the effect of both loci on lifespan is additive (Fig. 3j and k).

Across cohorts, the QTL on chromosome 18 explained 3.7% of phenotypic variance in lifespan and the QTL on chromosome 12 explained 2% of phenotypic variance. Collectively, the 4 QTL on chromosomes 2, 4, and 16 explained 5.2% of variance in lifespan. Together, loci detected in the mega-analysis explained 10.9% of overall trait variance and 58.6% of the estimated heritable variance for lifespan.

## Association mapping of genes underlying lifespan QTL

One strength of DO mice is the ability to convert allele probabilities at genotyped markers to genotype probabilities using the strain distribution patterns (SDPs) of individual variants in the 8 founding strains. Variant association mapping can then be performed on individual SNPs to further refine QTL (Methods). We performed variant association mapping of loci detected at the permutation-based significance threshold (Methods, Fig. 4) or at LOD ≥ 6 (Supplementary Fig. 3) in the mega-analysis (Supplementary Table 4). The 2LOD support intervals around the chromosome 18 loci in the Harrison and DRiDO studies encompassed ~26 protein coding genes (Supplementary Table 4). Variant association mapping further refined the effect of this locus to 10 genes: B4galt6, Ttr, Trappc8, Rnf125, Rnf138, Mep1b, Garem1, Klhl14, Ccdc178, and Axl3 (Fig. 4a). Association mapping of the chromosome 12 locus localized its effect to a region encompassing ~18 genes, including DRAGO/Susd6 (Polato *et al.* 2014), a DNA damage-responsive tumor suppressor and Rgs6, a regulator of G-protein signaling (Fig. 4b). Association mapping of the chromosome 16 locus localized its effect to ~8 genes: Rbfox1, Tmm114, Mettl22, Tmem186, Pmm2, Abat, Carhsp1, and Usp7 (Fig. 4c).

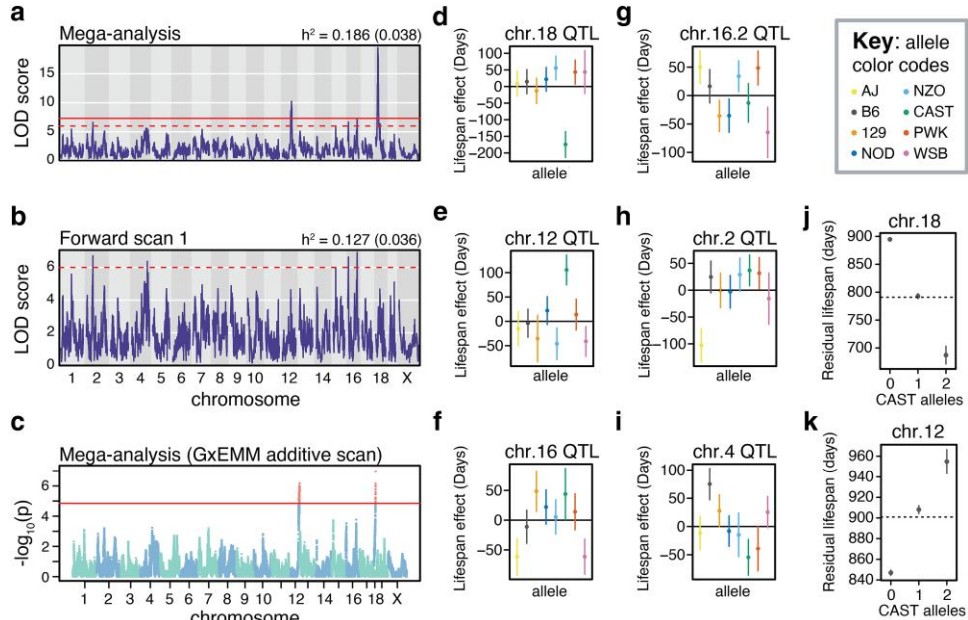

**Fig. 3.** Mega-analysis of lifespan in DO mice. a) QTL profile of the additive genome-wide scan on lifespan using combined data from the Harrison, Shock, and DRiDO studies. The solid horizontal line indicates genome-wide significance, while the dashed horizontal line indicates a significance threshold of LOD ≥ 6. The narrow-sense heritability ($h^2$) of lifespan is reported, with standard error about the $h^2$ estimate reported in parentheses (*top right*). b) QTL profile of the forward regression scan on the combined data, in which statistically significant loci were included as additive covariates in the model. c) Manhattan plot of the additive genome-wide scan on combined lifespan data using GxEMM. Markers surpassing the horizontal line are considered statistically significant. d–i) Allelic effects (BLUPs) and corresponding standard errors of d) the chromosome 18 locus, e) the chromosome 12 locus, f) the chromosome 16 locus previously detected in the Shock study, g) the second chromosome 16 locus, h) the chromosome 2 locus i) the chromosome 4 locus detected in the forward regression. j) Mean residual lifespan and standard error of DO mice as a function of the number of *CAST* alleles at the chromosome 18 locus. The dotted black line denotes the expected residual lifespan of individuals carrying a single *CAST* allele at this locus. k) Mean residual lifespan and standard error of DO mice as a function of the number of *CAST* alleles at the chromosome 12 locus. The dotted black line denotes the expected residual lifespan of individuals carrying a single *CAST* allele at this locus.

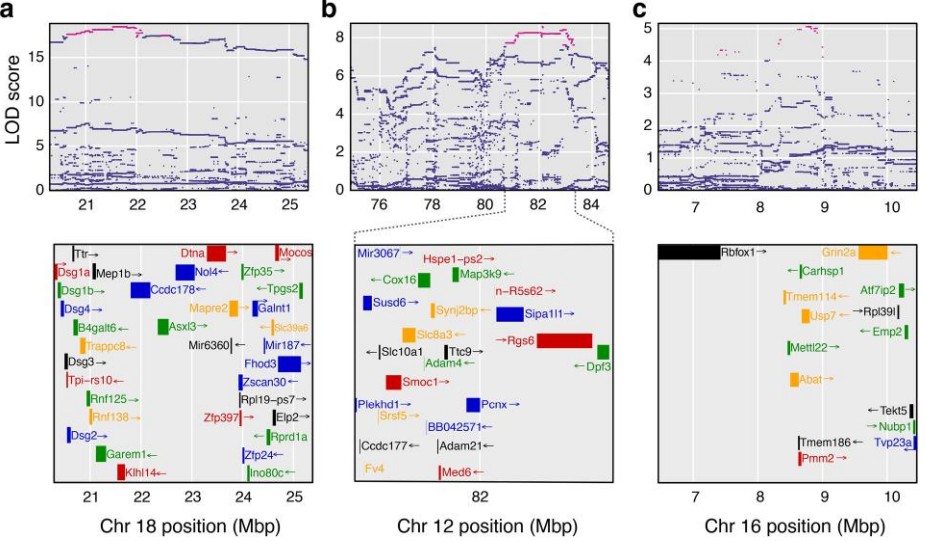

**Fig. 4.** Variant association mapping of loci detected at statistical significance in one or more studies. a) Variant association mapping of the QTL on chromosome 18 depicting the LOD scores for each variant within the 3 LOD support interval around the locus in the mega-analysis (*top*). The area encompassed by the plot corresponds to the 3 LOD support interval around the locus in the mega-analysis. The most likely candidate SNPs, defined as those with an LOD score within 1 LOD of the maximum variant association, are highlighted in pink. Genes within the genomic interval are depicted (*bottom*). b) Variant association mapping of the chromosome 12 locus in the mega-analysis. Here, the *bottom* panel is zoomed on the coordinates of the most likely causal variants for clarity (dashed lines). c) Variant association mapping of the chromosome 16.1 locus in the mega-analysis.

## Loci interact with sex and dietary restriction to influence lifespan in DO mice

Four of the 5 unique QTL identified in the individual studies only emerged in a single study. We evaluated the potential relevance of these 5 QTL across studies by fitting single-QTL models at the peak QTL positions using data from their respective studies, including interaction terms for sex (Shock study) or dietary intervention (DRiDO and Harrison studies). While none of the QTL

from the DRiDO or Harrison studies were found to have interactions with dietary intervention, the locus on chromosome 16 had a sex-specific effect on lifespan in the Shock cohort (single locus $\alpha = 0.05$, 1,000 permutations; Fig. 5a; Supplementary Table 5). The largest effects on lifespan at this locus were observed in female animals: after correcting for sex and DO generation, female mice carrying at least one copy of the PWK allele lived 150 days (SE = 18.73) longer than males, with the allele having a minimal effect on lifespan in males. None of the QTLs were found to have interactions with sex or dietary intervention in the mega-analysis ($\alpha = 0.05$, 1,000 permutations).

Motivated by the identification of a sex-responsive lifespan QTL, we conducted whole-genome meta analyses using gene-by-environment mixed models (GxEMM) to detect loci with treatment and/or sex-dependent effects on lifespan (Wright et al. 2022). Whole-genome scans for loci with diet-specific or sex-specific effects on lifespan resulted in detection of a single diet-responsive locus on Chromosome 5 at 112.74 Mb at a nominal $P < 10^{-4}$ (2LOD SI = 108.70, 115.39, Benjamini–Hochberg adjusted $P = 0.126$, Fig. 5b, supplementary Fig. 4). No sex-responsive loci were detected at a genome-wide significance threshold (Fig. 5c). The diet-responsive locus exhibited the strongest effects on lifespan under 40% CR, with the largest effects driven by alleles from the AJ and WSB founding strains (Fig. 5d). The 2LOD support interval about this locus spanned 3.78 megabases and contained 44 protein coding genes, limiting our ability to identify an obvious candidate (Fig. 5e, Supplementary Table 4).

## Discussion

Here, we performed genome-wide genetic analysis of lifespan in DO mice, combining data from 3 studies, 2 of which were previously unpublished, for a well-powered mega-analysis of 2,444 animals. These studies included diet and drug interventions, allowing us to evaluate those experimental interventions across a genetically diverse background and also evaluate the interactions between lifespan genetics and those environmental variables. Six QTLs significantly affected lifespan independently of diet and drug, one of which had a sex-dependent effect and another of which had a diet-specific effect on lifespan. The combined effects of these QTL could explain the majority of the estimated genetic variance for lifespan across these studies and thus provided a useful overview of the genetic architecture of lifespan in DO mice.

While interventions like CR have been shown to extend lifespan in organisms ranging from yeast to mice (Taormina and Mirisola 2014), the effects of these interventions may differ across genetic backgrounds within species (Liao et al. 2010). To address this, aging intervention studies such as the Intervention Testing Program (Miller et al. 2007) often repeat experiments in different contexts and/or genetic backgrounds to ensure that the effects of longevity treatments are robust. The studies described here all used DO mice and were conducted at Jackson Laboratory—albeit in different mouse rooms, at different times, and with some variation in protocols and housing conditions. Differences were observed between studies in the median lifespans of *ad lib* females, with median lifespan varying by approximately 16% between the DRiDO and Harrison studies. Environmental factors appeared to influence the variances of lifespan distributions, which in turn may have modified $h^2$ across the studies.

Lifespan variation between studies can be attributed in part to differences in the experimental designs of those studies; for example, mice in the DRiDO study were subject to intensive

phenotyping pipelines that captured hundreds of additional phenotypes, which may have impacted the lifespan of the animals, while animals in the Harrison study experienced minimal handling, were housed at a different temperature, and were subject to an automated 24 h feeding schedule. Additional sources of variation may include relatively small differences in facilities, personnel, phenotyping, diet composition (e.g. non-irradiated 5LG6 vs 5K52 chow), and co-housing, all of which may influence lifespan directly or through interaction with interventions (Huang et al. 2020; Zhao et al. 2022; Luciano and Churchill 2024). For instance, mice in the Harrison study were fed non-irradiated 5LG6 chow, which differs in both micronutrient composition and fat content from the 5K52 chow used in the other studies. Previous literature suggests that diet affects many health-related outcomes in mice, including body composition, cardiovascular traits, and immune composition (Zhang et al. 2022; Di Francesco et al. 2024). Experimental factors seem to have played a large role in influencing male lifespan, which had a $h^2$ indistinguishable from 0 and may have been affected by the co-housing of animals.

The influence of experimental variation in estimating intervention effects on aging is exacerbated by challenges associated with fitting Gompertzian models for lifespan, which assume homogenous populations with constant aging rates and can be skewed by mortality rates earlier and later in life (Easton 2009). Further, the baseline hazard associated with a lifespan intervention will have repercussions on the estimated mortality doubling time associated with said intervention (e.g. an elevated baseline hazard coupled with an extended lifespan may result in an artificially high MDT estimate, and vice versa). This may explain the effects of CR, rapamycin, and sex on MDT in the non-DRiDO studies, which appear correlated with the effects of the same variables on baseline hazard. Nonetheless, differences among these studies suggest that lifespan is labile to variations in experimental conditions and warrants caution when generalizing to even broader contexts.

Despite these differences, dietary interventions and rapamycin treatment increased lifespan across many genetically unique individuals in all studies (Fig. 1), with 30% CR extending lifespan by approximately 20% in the Harrison and Svenson studies, similar to the effects of 20% CR in the DRiDO study (Di Francesco et al. 2024). The hazard ratios associated with CR and rapamycin did not significantly differ between the Harrison and Svenson studies, further suggesting the effects of these interventions are relatively consistent despite varied experimental contexts.

Lifespan is an integrative phenotype that encapsulates many causes of mortality, each of which might have distinct genetic drivers. This property suggests that lifespan should therefore be a highly polygenic trait, making it difficult for any one genetic driver to explain a large fraction of its genetic variance. It is also a low heritability trait, with a substantial portion of variance attributable to the random chance associated with the exact timing of death events. Power to detect QTL is therefore limited, with a small amount of genetic variance likely diluted across a large number of loci. Additionally, each mortal trigger may have its own spectrum of gene-by-environment interactions, further diluting the power of genetic associations in the context of mega-analyses—and particularly relevant in our study due to the inclusion of dietary interventions, whose effects on health-related outcomes are known to vary across genetic backgrounds (Liao et al. 2010; Roy et al. 2021; Li et al. 2023). Despite such challenges to statistical power, 8 QTL influencing lifespan in DO mice were detected in our study. Of these, only 3 achieved genome-wide statistical significance in at least 2 of our analyses (including individual

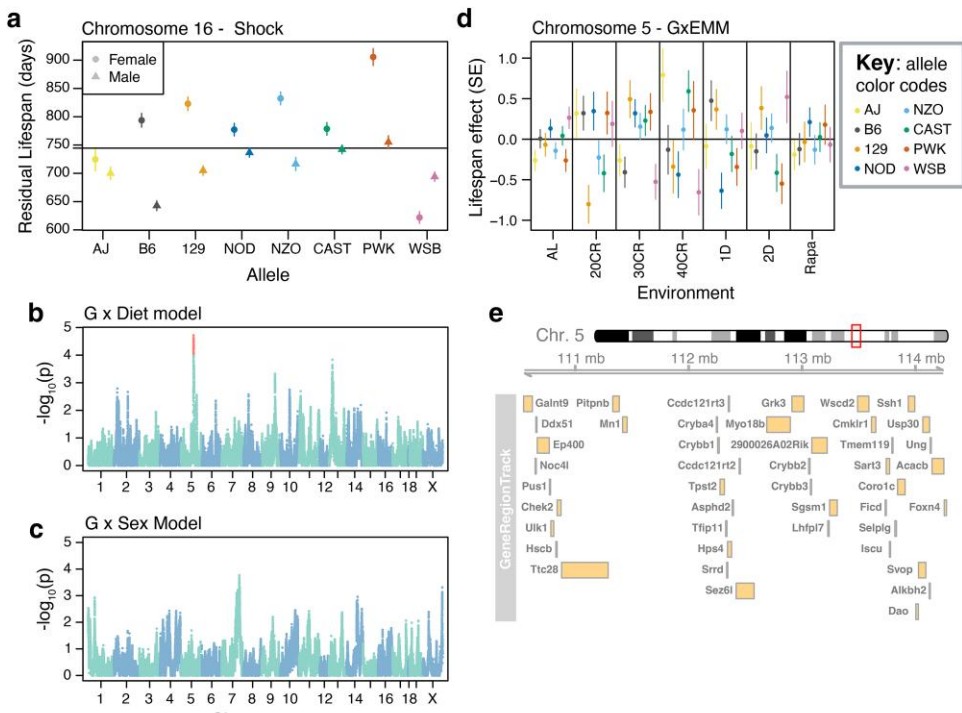

**Fig. 5.** Genome-wide scans for loci with diet- or sex-specific effects on lifespan. a) Sex-specific effects of the chromosome 16 locus within the Shock study. Mean lifespan after correcting for sex and generation wave is shown by the black line. Error bars denote standard error. b) Whole genome scan for loci influencing lifespan via interaction with diet using GxEMM. Red points denote loci with significant effects at $P < 1 \times 10^{-4}$. c) Whole genome scan for loci influencing lifespan via interaction with sex using GxEMM. d) Allelic effects of the chromosome 5 locus on lifespan as a function of dietary intervention or drug treatment, with standard errors. e) Map of the 2LOD support interval about the peak marker at chromosome 5 depicting all protein coding genes within the interval.

studies and the mega-analysis), highlighting the relevance of the aforementioned challenges. However, reproducible detection of 3 of the loci (chromosomes 12, 16.1, and 18) in multiple individual studies/mega-analysis suggested robust effects on physiology, likely impacting multiple facets of aging or disease.

Further exploration of these lifespan-influencing loci, particularly those that were most robust, is warranted. Pathophysiology of the DO founding strains may provide clues regarding the physiological processes affected by lifespan QTL, although this information is not available for all strains. Mortality in C57BL/6 mice, among the most common strains in the literature, is frequently driven by lymphoma and hematopoietic neoplasms (Brayton *et al.* 2012). The chromosome 18 locus was previously co-localized with a QTL for red blood cell distribution width (RDW; Di Francesco *et al.* 2024), which is a prognostic marker for several cancers of the blood (Herraez *et al.* 2020; Carlisi *et al.* 2024). Among the modest number of genes identified via association mapping at chromosome 18 are Rnf125 and Klhl14, tumor suppressors that are each involved in several cancers (Kodama *et al.* 2022; Draškovič *et al.* 2024), including Klhl14 in B-cell lymphoma (Choi *et al.* 2020). Given its role as a tumor suppressor, DRAGO/ Susd6 on chromosome 12 is a strong candidate. Garem1 emerges as an additional candidate on chromosome 18 due to its influence on body weight, a trait correlated with lifespan (Roy *et al.* 2021) and through which previously identified lifespan QTL have also been shown to impact longevity (Bou Sleiman *et al.* 2022; Nishino *et al.* 2022). In addition to lifespan, hundreds of additional phenotypes were measured in the DRiDO study, which may hint at alternative mechanisms underlying this lifespan effect. While the application of these phenotypes to loci detected in other lifespan studies is less apparent, follow-up analyses and experiments aimed at

identifying causal variants and the mechanisms by which they influence lifespan are warranted.

This study aimed to broadly define the genetic architecture of lifespan in DO mouse populations. Most analyses explored additive genetics, and the designs of component studies enabled meaningful gene-by-environment (GxE) analysis across diet and drug treatment groups. Less power was available for exploration of other components of genetic architecture, such as dominance and epistasis. Limited power was available for the exploration of gene-by-sex (GxSex) interactions, which have been more thoroughly explored in UM-HET3 mice (Bou Sleiman *et al.* 2022). A locus on chromosome 16 exhibited a GxSex effect, but more male animals would have been necessary for a thorough review of GxSex.

Our survey of the additive genetic landscape of lifespan was more comprehensive, with the detected QTL accounting for over half of the estimated heritability. As discussed above, the polygenicity for this trait is expected to be high, and the 6 loci detected in the mega analysis likely represent the high end of a long list of contributing QTL, with a long tail of diminishing effect sizes. Supposing that undetected QTL should explain less variance than the smallest identified QTL (1.3%), we project that there should be at least 13 loci with additive effects on DO mouse lifespan. Anticipating a long tail of small effect sizes, we expect the true number of QTL to be substantially higher than that projected minimum. However, with diminished effect sizes, the likelihood of broad physiological impact also diminishes. We therefore expect the loci described herein to deliver substantial insight into the processes that drive aging and mortality in mice, and we look forward to future interrogation of their mechanisms of action.

## Data availability

All Supplementary Tables and Data, which includes primary genetic and phenotype data, are available at Dryad: https://doi.org/10.5061/dryad.pnvx0k6z8.

Supplemental material available at GENETICS online.

## Acknowledgments

We would like to thank Nick Van Bruggen and Qingbo Wang for their thoughtful feedback on this manuscript.

## Funding

This work was funded by National Institutes of Health grants AG038070, AG022308, and AG079753, the Ellison Medical Foundation (G.A.C.), and by Calico Life Sciences LLC.

## Conflicts of interest

This work was partially funded by Calico Life Sciences LLC. M.N.M., K.M.W., A.R., A.D.F., and J.G.R. were employees of Calico Life Sciences LLC at the time the study was conducted.

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
