## [Peer Review File · Genetics]

Analysis of lifespan across Diversity Outbred mouse studies identifies multiple longevity-associated loci

Martin Mullis, Kevin Wright, Anil Raj, Daniel Gatti, Peter Reifsnyder, Kevin Flurkey, Jonathan Archer, Laura Robinson, Andrea Di Francesco, Karen Svenson, Ron Korstanje, David Harrison, J. Graham Ruby, and Gary Churchill

NOTE: The reviews and decision letters are unedited and appear as submitted by the reviewers.

In extremely rare instances and as determined by a Senior Editor or the EIC, portions of a review may be redacted. If a review is signed, the reviewer has agreed to no longer remain anonymous.

The review history appears in chronological order.

Review Timeline:

Submission Date:	2024-12-03
Editorial Decision:	2025-01-13
Resubmission Received:	2025-03-18
Accepted:	2025-03-31

January 8, 2025

GENETICS-2024-307683

Analysis of lifespan across Diversity Outbred mouse studies identifies multiple longevity-associated loci

Dear Gary,

Three experts in the field have reviewed your manuscript, and I have read it as well. With revision, your manuscript is potentially suitable for publication in GENETICS. The reviewers have comments and concerns that need to be addressed in a revised manuscript. You can read their reviews at the end of this email.

Please could you pay particular attention to revising the Discussion, which reviewer 2 (point 4), and reviewer 3 (point d) highlighted as a concern.

We look forward to receiving your revised manuscript. Please let the editorial office know approximately how long you expect to need for revisions.

Upon resubmission, please include:

1. A clean version of your manuscript;
2. A marked version of your manuscript in which you highlight significant revisions carried out in response to the major points raised by the editor/reviewers (track changes is acceptable if preferred);
3. A detailed response to the editor's/reviewers' comments and to the concerns listed above. Please reference line numbers in this response to aid the editors.

Additionally, please ensure that your resubmission is formatted for GENETICS.

<https://academic.oup.com/genetics/pages/general-instructions>

Follow this link to submit the revised manuscript: Link Not Available

Sincerely,

Jonathan Flint
Associate Editor
GENETICS

Approved by:
Anthony Long
Senior Editor
GENETICS

Reviewer #1 :

This manuscript conducts a genetic meta-analysis of lifespan in DO mice, integrating data from multiple, independently conducted studies. By examining how dietary interventions, rapamycin treatment, and sex influence longevity, the authors identify multiple QTL and explore gene-by-environment interactions. The use of genetically diverse populations and robust statistical methods strengthens the biological relevance.

Major Comments:

1) Although the manuscript acknowledges differences in study design providing a clearer discussion of how these differences both enhance the generalizability of the findings and introduce potential limitations would strengthen the interpretation of the results.

2) The manuscript identifies several QTL associated with lifespan and narrows down candidate genes, but the discussion provides minimal context for why these genes are relevant to aging. Given that the focus is on locus discovery rather than functional validation, even brief references to known gene functions or related aging pathways would give readers a clearer understanding of the biological significance of these findings.

Minor comments:

- 1) Mentioning lifespan QTL studies in other model organisms (e.g., worms, flies) could highlight the evolutionary conservation of longevity pathways and enrich the introduction's context.
- 2) While well-known in aging research, rapamycin and dietary restriction may be unfamiliar interventions to some readers. A short primer on their known or hypothesized modes of action, and why they are of particular interest for lifespan studies, would provide helpful context.
- 3) The Discussion acknowledges variations in baseline hazard and mortality doubling times across studies and interventions. Adding a brief explanation of the assumptions behind the Gompertz model, why these parameters were chosen, and potential biological or methodological reasons for these inconsistencies would help readers better interpret the results.
- 4) The Shock study's heritability estimate (15% {plus minus} 15%) indicates substantial uncertainty. Acknowledging this large standard error and, if possible, referencing sample sizes or other contributing factors would clarify why this estimate is less precise.
- 5) Clarifying how the authors harmonized genotypes from MUGA, MegaMUGA, and GigaMUGA arrays (e.g., common markers, imputation) and ensured data compatibility would strengthen the presentation of the meta-analysis.
- 6) The authors performed a QC step by comparing genotype-predicted versus recorded coat colors. If possible, mentioning whether sex verification was also performed (e.g., checking sex-linked markers) would be nice to include.

Reviewer #2 :

This work is an important new member of a series of papers that are exploiting the mouse Diversity Outbred populations to explore the genetics and environmental modulation of lifespan and health span.

This is also an interesting study because it is a mega-analysis of diversity at another level -- diversity among laboratories and their vivaria. Handling this higher level GxE on top of the intentional GxE of dietary instrumental variables is hard work, but the result in this paper is many interesting findings a small yield of accurately delimited QTLs that will be as useful to molecular geroscientists as they are to those of us more interested in genetic architecture of lifespan.

Figure are exemplary!

What could be done to improve this submission to increase the clarity of the presentation and the ultimate impact of the work? Here are some possibilities:

1. The term "genetic architecture" is a fuzzy beast that needs taming, whether in the introduction or in the discussion. Mapping QTLs--the focus of most of this study--is only a small part of "genetic architecture". The statistical geneticists reading the paper will recognize the patina of the theme throughout, but given that the phrase is highlighted in the first and last sentences of the abstract and in the final paragraph of the introduction, it definitely deserves a sharp-edge operational definition for most readers.
2. The power of this study is all on the side of females. This fact needs to be fielded explicitly; perhaps toward the front of the results before the remarkable summary of lifespan data of the four female cohorts on an ad libitum diet. In four specific pathogen-free colonies of females, all set up within a few years of each other, the median lifespans vary from 765 to 891 (126 days or 4.1 months). What this says, loud and clear, is that the genetic architecture of lifespan is actually "genetic x environmental" architecture.
3. This paper was written by a highly talented team of statisticians, and that results in an blind spot in presentation style in which higher order statistical outcomes are presented without giving the reader the basics. One example: "...the mortality doubling time was slightly increased by CR ($p = 5.83 \times 10^{-4}$)..." Nowhere in this most interesting paragraph is the reader given a handle on the actual doubling time of the hazard ratio. Yes, readers can perhaps look through the figures and estimate the value, but why not just put it in the text? (And think of the poor AI systems that will have all of the p values but no notion of the fundamental values.) Adding this type of text will only lengthen the Results by a p value of 0.034.
4. Many readers will be impressed by the "within-megastudy" variability in heritabilities of lifespan, all in SPF vivaria and all using cohorts that are of good size--from 0.15 to 0.25. They probably do not differ significantly, but it is still a good lesson. Is there a way to use non-parametric methods to bootstrap or jack knife your way to a consensus h^2 for a "mess of SPF environments" with an error term at the end of this paragraph? Is the h^2 different between males and females?
5. The QTL analysis of the DO cohorts is handled as well as it can be handled, and the results are both significant and useful.

But more context on the genetic architecture--the pros and cons of the DO--would be helpful. The haplotypes inherited from the "wild" inbred Ur-parents of the DO have their expected strong impact on lifespan.

A paragraph added to the discussion could provide an overview of how unusually high levels of genetic diversity impact the analysis of GxE architecture. Do we get out an analytic comfort zone when fielding 50 million variants with MAFs above 10%? I honestly do not know the answer, but I do know that I smiled when I got to the estimate of number of effective factors modulating lifespan in the DO. My own personal estimate but not using Sewall Wright or other statistical methods is that "about 10% of the genome" or perhaps the Hitchhiker's Guide value of 42 -- rather than "at least 12" as given at the end of the discussion.

6. There is a marked difference in the level of scholarship between the introduction and the discussion that puzzled me. The introduction has a high level of scholarship whereas the discussion flashes by the highlights and does not do what I expected--consider the genetic architecture in comparison to those of the two other species with the richest data on lifespan--fruit flies and humans. Charlesworth, Rose, Mackay, and many others have thought long and deeply on the genetic architecture of longevity in *Drosophila* populations, and some of their work has GxE components. They also are comfortable considering the evolutionary context of aging. In contrast, this discussion was written without any evolutionary or life history context. Go ahead and expand the discussion and see if results fit in a broader context. You will find that many of the results in this DO study were presaged by *Drosophila* studies and to lesser degree studies of human longevity and lifespan loci.

7. The difference between haplotype mapping and association mapping needs a bit more explanation. This is a cool feature of the DO.

8. This is not really a meta-analysis. It is what is usually called a mega-analysis in genetics since you have access to all data from all studies are combining and integrating from roots to leaves.

9. A comment on "Sex-specific effects" and "diet-by-locus" interactions and their apparent modesty in this study: Getting at even the bare-bones basics of genetic architecture takes more statistical power than we usually can muster--if by architecture we mean estimation of higher-order and often non-linear interaction effects--starting with dominance, epistasis, sex and sex chromosomes, mitochondrial genomes and other parent-of-origin effects, the many GxE effects. The fact that DO is also so genetically diverse does not make this effort any easier (British understatement). The observation of minimal sex specificity of loci and minimal sex-by-locus interaction may be more of a power issue than a fundamental feature of the DO. The finding that you can explain a lot of the h^2 with a handful of QTLs might be a rejoinder, but it is one that some of us will not take too seriously given the scope of GxE even within vivaria at the same institution.

10. Minor: Rsg6. I think this should be Rgs6.

11. Minor from Introduction: "GxE effect on lifespan remain unexplored". Well, yes at the locus level but only if you ignore years of work in *Drosophila* ;-). What would Mackay and others say?

The statement is not quite true even in mice. Many studies have tested for GxE at the organismal level. Recent examples include a series of studies by S Roy, K Mozhui, EG Williams and others on the phenotypic, epigenomics, and multiomic impact of GxE (diet) on lifespan in the BXDs. But I certainly could agree that in vertebrates there is not much that has been resolved GxE to the level of loci, let alone genes or variants.

12. Minor from introduction. The paragraph that begins: 'Dozen of lifespan-associated QTL...' is a bit boring and misdirected. Yes, there is certainly a grain of truth here, but it misses several important points. Mapping is all about managing trade-offs. If one truly cared about "genetic architecture" than a 10-by-10 diallel cross of Collaborative Cross strains in five environments would be fabulous, but of course is far from ideal for mapping QTL. If one cares about power and boosting heritability by resampling, then recombinant inbred strains or F1s are a good way to go. If one cares about community multiomics data sharing, then RI strains are the way to go. RI strains do not have inherently low mapping precision. They do very well for oligogenic traits and even polygenic traits--this was the entire point of the Collaborative Cross as well as the expanded BXD family, much of which was made from advanced intercross progeny at G8 to G12. With "merely" 120 BXD RI strains and no replication (N of 1 per strain), one can achieve 2 LOD CIs that rival DO but with high power since MAF is about 0.5. See Sasani TA, Ashbrook DG et al. in *Nature* 2022 for a good example.

The final touch in this paragraph that made me laugh was reading about the (implicitly high) mapping precision of the UM-HET3 progeny. The mean 1.5 LOD confidence intervals of lifespan loci from the Bou Sleiman paper are about 35 Mb with 2500 animals. That is not precisely precise.

Reviewer: RW Williams

Reviewer #3 :

This manuscript describes phenotypic/genetic analysis of 4 large DO mouse lifespan experiments; one of which was recently published in Nature (DRiDO) along with 3 previously unpublished studies. The authors describe treatment/sex effects - each study examines either both sexes or at least 1 dietary treatment - execute QTL mapping study-by-study and across studies in meta-analyses, and find several QTL, including those with apparent sex/treatment-specific effects.

Conceptually and scientifically I thought the study was strong, and added to the existing literature on the genetics of aging/lifespan.

I have a bunch of text/clarity issues, all of which are pretty minor (see below under "Relatively Minor Comments"). But my main concern is that the Discussion is weak and needs significant work (see my point "d" below).

Comments:

(a) Methods last.

I am not sure the "methods last" format of the manuscript works well; the first 2 paragraphs of the Results section refer to the "Methods" section 4 times for further information, and in at least 1 case - the 1st instance - I felt the need to go to the Methods to read it. I would advocate for switching the order of material.

(b) Inconsistent effects of treatment on demographic aging.

Much of page 4 describes some differences across studies in hazard and mortality doubling time, with the point being that the dietary treatments consistently show lifespan extension, but that exactly how this is achieved, and the "shape" of the survival curves that are produced, are different across studies. In the "Discussion" I was anticipating some robust discussion of these differences. But all that is stated (end of para 1 of the Discussion) is "...were less consistent, highlighting the difficulty of modeling Gompertzian mortality parameters." So - in my mind - this renders the Results section of page 4 of limited utility; if one cannot model these parameters well (as the discussion claims), then _is there actual_ inconsistency across studies? Or is there just not enough power/resolution/accuracy to properly determine consistency? I think these segments of the manuscript need work to clarify what the reader is supposed to take from them.

(c) Variant association mapping. The idea is to focus in on QTL windows and execute variant-by-variant "local" GWAS to help refine QTL intervals. But it isn't clear what statistical threshold is being used to do this. The "Association mapping of genes underlying lifespan QTL" section in the Results" doesn't describe this. Figure 4 legend states that the "most likely candidate SNPs" are highlighted in pink, but not what "most likely" means. The "Variant Association" methods section has a description of a permutation test, but this seems specific to interaction terms. More/better detail would assist interpretation here.

(d) The Discussion section is the weakest aspect of the manuscript; it doesn't really discuss the data/studies well, is very nebulous, and is poorly backed up with citations. In addition to my point "b" above, the discussion - in my read - basically suggests that some of the differences in the genetic results across studies are due to differences in design across studies, and some are due to GxTreatment. There is more to it as written, but not much. This was a weakness for me. The authors allude to maybe body weight being involved in a particular QTL. Since it looks like this was measured in the DRiDO study at least, perhaps this could be examined or at least more broadly discussed? Indeed, the DRiDO study includes a ton of other phenotypes measured; the Chr18 lifespan QTL reported here was previously reported in the DRiDO study where it overlaps with a blood QTL. Do the other QTL presented in the current manuscript overlap with non-lifespan QTL seen previously, including in the DRiDO study? Examining this would connect the present study better to the extant literature (and would also connect the Discussion to the 1st para in the Intro which is about the idea that "lifespan" is a trait with many underlying contributors). There is no discussion at all about the candidate genes; have any been associated with lifespan-associated previously traits? Are there human orthologs of interest? There is also no discussion about the QTL; in the many previous lifespan mouse studies has anyone mapped loci in these regions before? This would provide additional evidence for QTL that didn't quite reach genomewide significance, and perhaps support - depending on the founder strains used in prior studies - the haplotype effects observed here. Ultimately I think there are many ways the authors could go to enhance the discussion, and I'd be comfortable with many routes. But as it is the section needs a bunch of work.

Relatively Minor Comments

(No page or line numbers in manuscript, so I've pasted in sections from the text below so the authors can find the places in the text where my comments are directed).

"Lifespan is a quantitative phenotype that can..."

I like the thrust of this para, but there are a few edits I'd suggesting making to make it an easier read. (1) The word "correlates" is

used a few times. What is meant - I think - is "phenotypes that correlate with lifespan", but its assumed the reader will understand this. I think it should be explicit. (2) The term "the phenotype" or "this phenotype" is used a few times, meaning - I assume - "lifespan". Replacing the general term with the specific one would help. (3) On the 7th line of the para I think "sub-population or" is redundant. (4) On the 4th line I think "unrelated" means that the potential underlying components of lifespan are unrelated _to each other_. But one read is that one is talking about components that are unrelated to lifespan.

"The limited sample sizes of typical mouse experiments..."
Explicit # and citations would be useful here.

"lack of substantial population structure"
But human studies and DO studies typically account for this in the analysis, so is this a major difference?

"gene-by-environment interactions; GxE"
"genome-wide genetics-by-environment (GxE)"
(On page 2). Would perhaps be useful to be consistent in terminology.

"Those designs are summarized in Figure 1B-E..."
The figures don't show the designs, really just the # of animals in each treatment group

"Despite varied designs, one control group..."
Paragraph seems a very windy way to say the studies are different. Maybe could be a lot shorter.

"probed the effects of 30% caloric restriction ('30CR')"
The manuscript flips between DR/dietary and CR/caloric throughout. I would pick just 1 term and be consistent.

"Despite beginning treatment at 16 months of age...possibly because treatment began in midlife"
I would advocate the text here describe the figure better; the point appears to be that there is no AL versus Rapa difference until approx the median lifespan, but then after that point surviving mice on Rapa are longer lived.

"Caloric restriction at 30% also increased..."
In this paragraph basically the exact same words are used in 2 consecutive sentences to describe 2 of the studies. Feels like the text could be tidier/shorter.

"This result was reminiscent of the slight female survival advantage..."
The statement here sort of implies the current work replicates a niche result seen a couple of times previously. But isn't it the case that across _lots_ of systems females tend to live longer?

"We estimated both the rates of demographic aging..."
This para states that "mortality doubling time was slightly increased by CR" in Harrison (but in 1F the value goes down not up) and that "caloric restriction significantly reduced the mortality doubling time" in DRiDO (but in 1F the value goes up and not down as implied).

"In the Shock study, neither baseline hazard..."
This single sentence isn't really a paragraph, and perhaps could be incorporated elsewhere.

"Notably, this locus was not detected at genome-wide significance when excluding mice that did not survive until the initiation of dietary intervention (6 months)."
A couple of points. First, in 1D, it looks like the % survival data might only start at 6 months, so from the figure it doesn't seem like any animals died before diets were changed. The DRiDO study suggests 23 did die; maybe this would be worth stating. Second, when doing the analysis with the full set of 960 mice and finding the QTL above threshold, how where the 23 mice that never received a dietary treatment "coded"? AL ? Third, the way the sentence is presented suggests this is interesting in some biological way; are the authors suggesting there is something interesting about the genetic composition of the 23 pre-6 month deaths? They presumably could examine this. Fourth, is it plausible this is just a power issue? (If you dump any 23 animals does the QTL fail to reach significance.)

"at a previously established significance threshold"
How was it previously established? How is it clear that the value translates to the present study?

"Assuming that remaining additive effects explain less phenotypic variance..."
I would move this to the discussion (there is similar material there. Although I would tend to advocate removing it all since it makes a lot of assumptions about the distribution of the un-identified / unknown QTL effect sizes.)

"including interaction terms for sex (Shock study) or dietary intervention (DRiDO study)."
The Harrison study also has interaction terms, but is not re-tested?

"None of the QTL were found to have interactions with sex or dietary..."
Is this perhaps just a power issue?

"the detection of QTL unique to individual studies raises the possibility that differences in experimental design..."
Also just statistical power could explain this?

"different dietary interventions have been shown to have contrasting effects on health-related phenotypes"
Citations?

"This QTL had the largest effect..."
Which QTL is being referred to is not 100% clear.

"roughly estimate that at least 12 QTL would be required"
Sure. But it could equally be 1200, right?

Figure 1

- Would be useful to describe better in the figure legend the set up for the 4 studies (e.g. that B/C/D are all females).
- Mouse numbers in 1C x-axis legend are wrong (same as 1B)
- All the abbreviations need to be in the legend (e.g. "2D") so the figure can stand alone.
- Legend states "Vertical bars on each curve indicate censorship events" Not seeing these.

Figure 2

- A-C are not really Manhattan plots; they're QTL profiles.
- The grayed-out square in the lower right of 2E looks a bit odd.
- The legend for 2D implies the Chr18 QTL was detected in the Shock study "detected in the Harrison (left), DRiDO (middle), or Shock (left) study"

Figure 3

- A-B are not really Manhattan plots; they're QTL profiles.
- Legends to D-I are very repetitive.
- Particularly in J the tiny horizontal black line looks like part of the "1" CAST allele symbol. I would advocate for a dotted line through the entire of the middle of the plot to indicate where 'additive' would be.

Figure 4

- As mentioned above, it would be useful to define in the legend what "most likely" means statistically.
- Is there anything meaningful to the gene color-coding scheme? Would perhaps help with "seeing" the candidates if those were colored differently than the non-candidates (within QTL but not highlighted by association) and than those not in the QTL (outside the 2-LOD QTL drop).

Figure 5

- D needs to be made wider to properly see the bars.
- E is not readable at this resolution. Either make much bigger, or delete, or make bigger and put into supplement.

"Dietary Restriction (DRiDO):"
Not clear what food type is here; just states "g" but not grams of what.

"Additive whole-genome scans:"
Not including kinship in the analyses? This would be atypical for DO QTL scans, no?

"While less conservative, this threshold is more stringent than a previously reported method"
This is a little nebulous. More concrete detail would be useful.

Associate Editor Comments:

Reviewer #1 :

This manuscript conducts a genetic meta-analysis of lifespan in DO mice, integrating data from multiple, independently conducted studies. By examining how dietary interventions, rapamycin treatment, and sex influence longevity, the authors identify multiple QTL and explore gene-by-environment interactions. The use of genetically diverse populations and robust statistical methods strengthens the biological relevance.

Major Comments:

1) Although the manuscript acknowledges differences in study design, providing a clearer discussion of how these differences both enhance the generalizability of the findings and introduce potential limitations would strengthen the interpretation of the results.

The discussion of the manuscript has been significantly revamped with this statement in mind. Specifically, we have expanded upon how the differences among studies highlights the robustness of the interventions, while also acknowledging that differences in experimental design may impact lifespan outcomes across study or even cause the effects of the interventions to differ.

Later, we discuss how several QTL fail to reproduce across study or in the meta-analysis, and how this may be due to experimental differences, lack of statistical power, or both.

2) The manuscript identifies several QTL associated with lifespan and narrows down candidate genes, but the discussion provides minimal context for why these genes are relevant to aging. Given that the focus is on locus discovery rather than functional validation, even brief references to known gene functions or related aging pathways would give readers a clearer understanding of the biological significance of these findings.

Potentially any gene could influence lifespan, which makes predicting causal genes difficult even when the confidence intervals of lifespan QTL are fairly resolved. The experimental context is also important to consider, since different conditions will impact the risk associated with various causes of mortality and the genetic factors associated with mortality will shift accordingly.

Nonetheless, we've identified several candidates that appear more plausible for several of the QTL (chromosomes 12 and 18) based on the pathophysiology of C57BL/6 mice and studies linking bodyweight to lifespan. This has been added to the discussion of the manuscript.

Minor comments:

1) Mentioning lifespan QTL studies in other model organisms (e.g., worms, flies) could highlight the evolutionary conservation of longevity pathways and enrich the introduction's context.

We've added some citations to the introduction that point to papers mapping lifespan QTL in species ranging from yeast to mice. Rather than get into functional coherence among lifespan QTL across species, we attempt to better discuss how experimental context can influence effects of various loci on lifespan in the discussion of the manuscript.

2) While well-known in aging research, rapamycin and dietary restriction may be unfamiliar interventions to some readers. A short primer on their known or hypothesized modes of action, and why they are of particular interest for lifespan studies, would provide helpful context.

We've added a bit of text here to provide some background for both of these interventions, noting that they have been shown to extend lifespan across a range of organisms and appear to act on metabolic and nutrient-sensing pathways (although direct mechanisms of action on physiology are not known).

3) The Discussion acknowledges variations in baseline hazard and mortality doubling times across studies and interventions. Adding a brief explanation of the assumptions behind the Gompertz model, why these parameters were chosen, and potential biological or methodological reasons for these inconsistencies would help readers better interpret the results.

Gompertz models assume homogeneous populations and constant aging rates, and as a result often fit early- and late-life mortality poorly (e.g. due to survivorship bias or different mortality causes/rates at different ages). In addition, model fitting is sensitive to early mortality events, which may influence hazard estimates and have repercussions for the estimation of aging rates. We now directly address these assumptions and challenges in the discussion, pointing out that inconsistencies in the effects of interventions on aging rate may be driven by technicalities of model fitting in addition to experimental differences.

4) The Shock study's heritability estimate (15% {plus minus} 15%) indicates substantial uncertainty. Acknowledging this large standard error and, if possible, referencing sample sizes or other contributing factors would clarify why this estimate is less precise.

The Shock study avoided the use of sibling pairs and as a result the individuals in this study are all roughly equally related to one another - resulting in a "flat" kinship matrix. This may result in difficulties estimating h^2 within this study, as the range of kinship used to support the estimate is limited. We've made note of this in the main text.

5) Clarifying how the authors harmonized genotypes from MUGA, MegaMUGA, and GigaMUGA arrays (e.g., common markers, imputation) and ensured data compatibility would strengthen the presentation of the meta-analysis.

The following text has been added to the "*Genotyping*" section of the methods:

"To facilitate meta-analysis across datasets sequenced using different MUGA arrays, genotype probabilities were first interpolated to a set of 69,005 pseudo-markers and then converted to haplotype probabilities via hidden Markov model in R/qt12."

6) The authors performed a QC step by comparing genotype-predicted versus recorded coat colors. If possible, mentioning whether sex verification was also performed (e.g., checking sex-linked markers) would be nice to include.

In the methods section describing the Shock study, we now note that a genotyping quality-assessment was performed using animal sex. This was only performed for the Shock study, as the other studies were all female.

Reviewer #2 :

This work is an important new member of a series of papers that are exploiting the mouse Diversity Outbred populations to explore the genetics and environmental modulation of lifespan and health span.

This is also an interesting study because it is a mega-analysis of diversity at another level -- diversity among laboratories and their vivaria. Handling this higher level GxE on top of the intentional GxE of dietary instrumental variables is hard work, but the result in this paper is many interesting findings a small yield of accurately delimited QTLs that will be as useful to molecular geroscientists as they are to those of us more interested in genetic architecture of lifespan.

Figures are exemplary!

What could be done to improve this submission to increase the clarity of the presentation and the ultimate impact of the work? Here are some possibilities:

1. The term "genetic architecture" is a fuzzy beast that needs taming, whether in the introduction or in the discussion. Mapping QTLs--the focus of most of this study--is only a small part of "genetic architecture". The statistical geneticists reading the paper will recognize the patina of the theme throughout, but given that the phrase is highlighted in the first and last sentences of the abstract and in the final paragraph of the introduction, it definitely deserves a sharp-edge operational definition for most readers.

Genetic architecture is multi-faceted, and we now acknowledge this in the revamped discussion section. We are predominantly exploring the additive effects on lifespan in this manuscript and utilizing multiple experiments to assess the robustness of this aspect of genetic architecture. However, we can make several predictions about the genetic architecture of lifespan based on what effects were (or were not) detected using this relatively large cohort of DO mice. For example, the apparently minimal contribution of GxE and GxSex effects suggests that these effects (which are likely prevalent) each explain small fractions of lifespan variation in the experimental contexts of this manuscript.

The above points have been added to the discussion of the paper, and are relevant to Reviewer #2's points 5, 6, and 9 regarding GxE, GxSex, genetic architecture, and the broader context of genetic analysis of lifespan.

2. The power of this study is all on the side of females. This fact needs to be fielded explicitly; perhaps toward the front of the results before the remarkable summary of lifespan data of the four female cohorts on an ad libitum diet. In four specific pathogen-free colonies of females, all set up within a few years of each other, the median lifespans vary from 765 to 891 (126 days or 4.1 months). What this says, loud and clear, is that the genetic architecture of lifespan is actually "genetic x environmental" architecture.

The following text was added to the beginning of the results section:

“Male mice (n = 240) were only included in the Shock study, which examined lifespan in both sexes and did not include any interventions. The vast majority of lifespan data collected in these studies comes from female mice (n = 2,912) on either an *ad libitum* diet, one of several dietary interventions, or a rapamycin-supplemented diet.”

Additionally, the discussion section of the paper has been reworked and better acknowledges the role of the environment on the genetics of lifespan as well as lifespan itself (there may be independent effects of the environment on lifespan itself and the genetics of lifespan).

3. This paper was written by a highly talented team of statisticians, and that results in an blind spot in presentation style in which higher order statistical outcomes are presented without giving the reader the basics. One example: "...the mortality doubling time was slightly increased by CR ($p = 5.83 \times 10^{-4}$)..." Nowhere in this most interesting paragraph is the reader given a handle on the actual doubling time of the hazard ratio. Yes, readers can perhaps look through the figures and estimate the value, but why not just put it in the text? (And think of the poor AI systems that will have all of the p values but no notion of the fundamental values.) Adding this type of text will only lengthen the Results by a p value of 0.034.

We've amended the section of the paper discussing Gompertzian hazard models to include baseline hazard percentages and mortality doubling time (MDT) values in addition to the p-values. The text in the section has been reworked and a few errors have been corrected as well!

4. Many readers will be impressed by the "within-megastudy" variability in heritabilities of lifespan, all in SPF vivaria and all using cohorts that are of good size-from 0.15 to 0.25. They probably do not differ significantly, but it is still a good lesson. Is there a way to use non-parametric methods to bootstrap or jack knife your way to a consensus h^2 for a "mess of SPF environments" with an error term at the end of this paragraph? Is the h^2 different between males and females?

We report the h^2 of lifespan in the meta-analysis of the three studies (0.18, with a standard error of 0.038). This estimate is close to the mean h^2 of the individual studies and seems to be a good consensus. This estimate includes all lifespan measurements and accounts for study, sex, intervention, and DO generation.

We've added h^2 estimates for males and females to the main text.

5. The QTL analysis of the DO cohorts is handled as well as it can be handled, and the results are both significant and useful. But more context on the genetic architecture--the pros and cons of the DO--would be helpful. The haplotypes inherited from the "wild" inbred Ur-parents of the DO have their expected strong impact on lifespan.

A paragraph added to the discussion could provide an overview of how unusually high levels of genetic diversity impact the analysis of GxE architecture. Do we get out of an analytic comfort zone when fielding 50 million variants with MAFs above 10%? I honestly do not know the answer, but I do know that I smiled when I got to the estimate of the number of effective factors modulating lifespan in the DO. My own personal estimate but not using Sewall Wright or other statistical methods is that

"about 10% of the genome" or perhaps the Hitchhiker's Guide value of 42 -- rather than "at least 12" as given at the end of the discussion.

While our estimate of the number of factors associated with lifespan is the best we can do based on the h^2 and effect sizes of lifespan QTL, we now acknowledge that the true number of variants modulating lifespan is likely to be much higher. We also make statements re: the effect sizes of GxE and GxSex effects in the data, which are likely to be prevalent despite their apparent absence in the data but must also have individually small effects on lifespan.

6. There is a marked difference in the level of scholarship between the introduction and the discussion that puzzled me. The introduction has a high level of scholarship whereas the discussion flashes by the highlights and does not do what I expected--consider the genetic architecture in comparison to those of the two other species with the richest data on lifespan--fruit flies and humans. Charlesworth, Rose, Mackay, and many others have thought long and deeply on the genetic architecture of longevity in *Drosophila* populations, and some of their work has GxE components. They also are comfortable considering the evolutionary context of aging. In contrast, this discussion was written without any evolutionary or life history context. Go ahead and expand the discussion and see if results fit in a broader context. You will find that many of the results in this DO study were presaged by *Drosophila* studies and to a lesser degree studies of human longevity and lifespan loci.

The discussion section has been entirely revamped to include additional discussion of lifespan analysis in other systems, including *drosophila*.

7. The difference between haplotype mapping and association mapping needs a bit more explanation. This is a cool feature of the DO.

Agreed, this is a key strength of the system that was glossed over in the original manuscript. We've added some language to the variant association mapping sections (Methods and Main text) that touches on what association mapping is and how it is performed. We've also specified how we are defining the "most likely" causal variants in the association mapping.

8. This is not really a meta-analysis. It is what is usually called a mega-analysis in genetics since you have access to all data from all studies are combining and integrating from roots to leaves.

The meta-analysis in the paper is now referred to as a mega-analysis.

9. A comment on "Sex-specific effects" and "diet-by-locus" interactions and their apparent modesty in this study: Getting at even the bare-bones basics of genetic architecture takes more statistical power than we usually can muster--if by architecture we mean estimation of higher-order and often non-linear interaction effects--starting with dominance, epistasis, sex and sex chromosomes, mitochondrial genomes and other parent-of-origin effects, the many GxE effects. The fact that DO is also so genetically diverse does not make this effort any easier (British understatement). The observation of minimal sex specificity of loci and minimal sex-by-locus interaction may be more of a power issue than a fundamental feature of the DO. The finding that you can explain a lot of the h^2 with a handful of QTLs might be a rejoinder, but it is one that some of us will not take too seriously given the scope of GxE even within vivaria at the same institution.

We now acknowledge in the discussion that these are likely to exist and are likely prevalent based on how frequently they are detected in the literature. We also acknowledge that one reason for the apparent absence of such effects in this analysis is due to a lack of statistical power. However, we

can also say based on the effect sizes of the additive QTL and fraction of h^2 explained that GxE and GxSex effects, in the experimental contexts of this study, must be individually small.

10. Minor: Rsg6. I think this should be Rgs6.

Nice catch. Fixed.

11. Minor from Introduction: "GxE effect on lifespan remain unexplored". Well, yes at the locus level but only if you ignore years of work in *Drosophila* ;-). What would Mackay and others say?

The statement is not quite true even in mice. Many studies have tested for GxE at the organismal level. Recent examples include a series of studies by S Roy, K Mozhui, EG Williams and others on the phenotypic, epigenomics, and multiomic impact of GxE (diet) on lifespan in the BXDs. But I certainly could agree that in vertebrates there is not much that has been resolved GxE to the level of loci, let alone genes or variants.

We agree with both of the above points and did not intend to suggest that GxE effects on lifespan haven't been studied. In fact, these interactions have been studied extensively in many experimental systems. However, in the context of the interventions presented in this manuscript (rapamycin and caloric restriction), GxE effects do remain largely unexplored at the locus level. We've cleaned up this bit of the introduction... hopefully our point is presented more clearly now.

12. Minor from introduction. The paragraph that begins: 'Dozen of lifespan-associated QTL...' is a bit boring and misdirected. Yes, there is certainly a grain of truth here, but it misses several important points. Mapping is all about managing trade-offs. If one truly cared about "genetic architecture" than a 10-by-10 diallel cross of Collaborative Cross strains in five environments would be fabulous, but of course is far from ideal for mapping QTL. If one cares about power and boosting heritability by resampling, then recombinant inbred strains or F1s are a good way to go. If one cares about community multiomics data sharing, then RI strains are the way to go. RI strains do not have inherently low mapping precision. They do very well for oligogenic traits and even polygenic traits- this was the entire point of the Collaborative Cross as well as the expanded BXD family, much of which was made from advanced intercross progeny at G8 to G12. With "merely" 120 BXD RI strains and no replication (N of 1 per strain), one can achieve 2 LOD CIs that rival DO but with high power since MAF is about 0.5. See Sasani TA, Ashbrook DG et al. in Nature 2022 for a good example.

The final touch in this paragraph that made me laugh was reading about the (implicitly high) mapping precision of the UM-HET3 progeny. The mean 1.5 LOD confidence intervals of lifespan loci from the Bou Sleiman paper are about 35 Mb with 2500 animals. That is not precisely precise.

We've removed language suggesting that mapping resolution is poor in RI lines and higher in outbred populations.

Reviewer: RW Williams

Reviewer #3 :

This manuscript describes phenotypic/genetic analysis of 4 large DO mouse lifespan experiments; one of which was recently published in Nature (DRiDO) along with 3 previously unpublished studies. The authors describe treatment/sex effects - each study examines either both sexes or at least 1 dietary treatment - execute QTL mapping study-by-study and across studies in meta-analyses, and find several QTL, including those with apparent sex/treatment-specific effects.

Conceptually and scientifically I thought the study was strong, and added to the existing literature on the genetics of aging/lifespan.

I have a bunch of text/clarity issues, all of which are pretty minor (see below under "Relatively Minor Comments"). But my main concern is that the Discussion is weak and needs significant work (see my point "d" below).

Comments:

(a) Methods last.

I am not sure the "methods last" format of the manuscript works well; the first 2 paragraphs of the Results section refer to the "Methods" section 4 times for further information, and in at least 1 case - the 1st instance - I felt the need to go to the Methods to read it. I would advocate for switching the order of material.

The methods are now presented before the results in the manuscript.

(b) Inconsistent effects of treatment on demographic aging.

Much of page 4 describes some differences across studies in hazard and mortality doubling time, with the point being that the dietary treatments consistently show lifespan extension, but that exactly how this is achieved, and the "shape" of the survival curves that are produced, are different across studies. In the "Discussion" I was anticipating some robust discussion of these differences. But all that is stated (end of para 1 of the Discussion) is "...were less consistent, highlighting the difficulty of modeling Gompertzian mortality parameters." So - in my mind - this renders the Results section of page 4 of limited utility; if one cannot model these parameters well (as the discussion claims), then _is there actual_ inconsistency across studies? Or is there just not enough power/resolution/accuracy to properly determine consistency? I think these segments of the manuscript need work to clarify what the reader is supposed to take from them.

This is a great point. Part of our intention in writing this section was to compare the effects of interventions on aging rates and hazard to those reported in previous work, e.g. Di Francesco et al. in which caloric restriction slowed the rate of aging while minimally impacting hazard.

We found that the consistency of the interventions' lifespan effects did not translate into aging rate/hazard values. Due to the assumptions of Gompertz models (consistent aging rates, homogeneous populations, etc.) and challenges fitting the models (they can be skewed by early- and late-life mortality), we believe caution is warranted when generalizing results from Gompertzian models across study. Specifically, trends influencing baseline hazard may

influence the fit of the mortality doubling time (rate of aging) parameter and can change how one interprets lifespan effects.

The discussion section of the manuscript now touches on this more directly.

(c) Variant association mapping. The idea is to focus in on QTL windows and execute variant-by-variant "local" GWAS to help refine QTL intervals. But it isn't clear what statistical threshold is being used to do this. The "Association mapping of genes underlying lifespan QTL" section in the Results" doesn't describe this. Figure 4 legend states that the "most likely candidate SNPs" are highlighted in pink, but not what "most likely" means. The "Variant Association" methods section has a description of a permutation test, but this seems specific to interaction terms. More/better detail would assist interpretation here.

We've added some text to the "Association mapping" methods section that further describes the association mapping process (converting allele probabilities to genotype probabilities using strain distribution patterns for SNPs) and defines our threshold for determining the "most likely" candidate SNPs. In short, SNPs within 1LOD of the sentinel SNP in each QTL support interval are considered "most likely" candidates.

(d) The Discussion section is the weakest aspect of the manuscript; it doesn't really discuss the data/studies well, is very nebulous, and is poorly backed up with citations. In addition to my point "b" above, the discussion - in my read - basically suggests that some of the differences in the genetic results across studies are due to differences in design across studies, and some are due to GxTreatment. There is more to it as written, but not much. This was a weakness for me. The authors allude to maybe body weight being involved in a particular QTL. Since it looks like this was measured in the DRiDO study at least, perhaps this could be examined or at least more broadly discussed? Indeed, the DRiDO study includes a ton of other phenotypes measured; the Chr18 lifespan QTL reported here was previously reported in the DRiDO study where it overlaps with a blood QTL. Do the other QTL presented in the current manuscript overlap with non-lifespan QTL seen previously, including in the DRiDO study? Examining this would connect the present study better to the extant literature (and would also connect the Discussion to the 1st para in the Intro which is about the idea that "lifespan" is a trait with many underlying contributors). There is no discussion at all about the candidate genes; have any been associated with lifespan-associated previously traits? Are there human orthologs of interest? There is also no discussion about the QTL; in the many previous lifespan mouse studies has anyone mapped loci in these regions before? This would provide additional evidence for QTL that didn't quite reach genomewide significance, and perhaps support - depending on the founder strains used in prior studies - the haplotype effects observed here. Ultimately I think there are many ways the authors could go to enhance the discussion, and I'd be comfortable with many routes. But as it is, the section needs a bunch of work.

We agree, and have re-written the discussion section from scratch to include broader discussion of: genetic architecture, the impact of experimental differences on intervention effects and the genetics of lifespan, the challenges associated with fitting Gompertzian models, possible candidate genes underlying QTL, and previous work on lifespan genetics in mice and other systems.

Relatively Minor Comments

(No page or line numbers in manuscript, so I've pasted in sections from the text below so the authors can find the places in the text where my comments are directed).

"Lifespan is a quantitative phenotype that can..."

I like the thrust of this para, but there are a few edits I'd suggesting making to make it an easier read.

(1) The word "correlates" is used a few times. What is meant - I think - is "phenotypes that correlate with lifespan", but its assumed the reader will understand this. I think it should be explicit. (2) The term "the phenotype" or "this phenotype" is used a few times, meaning - I assume - "lifespan".

Replacing the general term with the specific one would help. (3) On the 7th line of the para I think "sub-population or" is redundant. (4) On the 4th line I think "unrelated" means that the potential underlying components of lifespan are unrelated _to each other_. But one read is that one is talking about components that are unrelated to lifespan.

- (1) We've rephrased "correlates" to "correlated factors" or just "factors" and now provide examples of what some of these factors might be (e.g. phenotypes, environmental effects, or genetic loci).
- (2) "The phenotype" is now explicitly stated: lifespan.
- (3) Here, we want to acknowledge that a factor influencing lifespan might influence certain aspects of health in an entire population, all aspects of health in a subset of a population, or some combination of the two. Subsets of a population and aspects of health are not redundant.
- (4) The reviewer's initial read here was correct; however, we agree that this phrasing was ambiguous. We've changed "unrelated" to "independent" to reflect that there may be many factors influencing lifespan through mechanisms independent of one another.

"The limited sample sizes of typical mouse experiments..."

Explicit # and citations would be useful here.

Added a few representative references and now state that sample sizes ranging from "hundreds to several thousand" are typical of DO mapping studies.

"lack of substantial population structure"

But human studies and DO studies typically account for this in the analysis, so is this a major difference?

I think so.

"gene-by-environment interactions; GxE"

"genome-wide genetics-by-environment (GxE)"

(On page 2). Would perhaps be useful to be consistent in terminology.

The "genome-wide genetics-by-environment (GxE)" scan is now referred to as a "genome-wide GxE" scan, since GxE is previously defined and that abbreviation is previously used.

"Those designs are summarized in Figure 1B-E..."

The figures don't show the designs, really just the # of animals in each treatment group

This sentence now reads:

"The treatment groups present in each study are summarized in the upper panels of **Figure 1B-E** and the corresponding experimental designs are described in more detail in the **Methods**."

"Despite varied designs, one control group..."

Paragraph seems a very windy way to say the studies are different. Maybe could be a lot shorter.

Our goal here is to capture the nuances of the differences among studies; not all sets of lifespan data are statistically different from one another.

"probed the effects of 30% caloric restriction ('30CR')"

The manuscript flips between DR/dietary and CR/caloric throughout. I would pick just 1 term and be consistent.

Dietary restriction (DR) is not always implemented via caloric restriction (CR), e.g. intermittent fasting, so there are points in the manuscript (for example, the introduction) in which DR is the more appropriate term to use. However, to avoid confusion, we now explicitly refer to dietary restriction and no longer use the acronym DR.

DR was used in several places in the methods where CR was the more appropriate/specific term. We've amended this.

"Despite beginning treatment at 16 months of age...possibly because treatment began in midlife"

I would advocate the text here describe the figure better; the point appears to be that there is no AL versus Rapa difference until approx the median lifespan, but then after that point surviving mice on Rapa are longer lived.

That paragraph has been rephrased to more clearly state that the effect of rapamycin on lifespan is driven by an increase in survival after 16 months, because that was when the treatment began.

"Caloric restriction at 30% also increased..."

In this paragraph basically the exact same words are used in 2 consecutive sentences to describe 2 of the studies. Feels like the text could be tidier/shorter.

This has been rephrased, and we used this as an opportunity to point out that 30CR had a similar effect on hazard ratios in the Harrison and Svenson studies despite the significant difference in lifespan between AL females from these studies.

"This result was reminiscent of the slight female survival advantage..."

The statement here sort of implies the current work replicates a niche result seen a couple of times previously. But isn't it the case that across _lots_ of systems females tend to live longer?

Yes, but what drives this isn't always clear. There are also a lot of systems in which males live longer. See Austad (2011) for details (now cited in the manuscript). Sex differences in longevity appear to vary by strain background, diet, housing conditions, temperature, and other factors.

"We estimated both the rates of demographic aging..."

This para states that "mortality doubling time was slightly increased by CR" in Harrison (but in 1F the value goes down not up) and that "caloric restriction significantly reduced the mortality doubling time" in DRiDO (but in 1F the value goes up and not down as implied).

Fixed this!

"In the Shock study, neither baseline hazard..."

This single sentence isn't really a paragraph, and perhaps could be incorporated elsewhere.

That sentence has been absorbed by the previous paragraph.

"Notably, this locus was not detected at genome-wide significance when excluding mice that did not survive until the initiation of dietary intervention (6 months)."

A couple of points. First, in 1D, it looks like the % survival data might only start at 6 months, so from the figure it doesn't seem like any animals died before diets were changed. The DRiDO study suggests 23 did die; maybe this would be worth stating. Second, when doing the analysis with the full set of 960 mice and finding the QTL above threshold, how were the 23 mice that never received a dietary treatment "coded"? AL? Third, the way the sentence is presented suggests this is interesting in some biological way; are the authors suggesting there is something interesting about the genetic composition of the 23 pre-6 month deaths? They presumably could examine this. Fourth, is it plausible this is just a power issue? (If you dump any 23 animals does the QTL fail to reach significance.)

This was unintentionally overstated; the significance of the locus is reduced but it still has an effect on lifespan. The original statement has been removed from the text.

Mice that failed to survive until dietary intervention were encoded as AL, and unlike the original DRiDO study, were included in our genetic analysis. However, to reproduce the previously reported survival data from the DRiDO, survival analysis in that study began at the time of dietary intervention (6 months). The text (see methods) now states this explicitly.

"at a previously established significance threshold"

How was it previously established? How is it clear that the value translates to the present study?

We've updated our significance thresholding for the GxEMM tests reported in Fig. 3 and Fig. 5 of the manuscript. For the additive scan (reported in Fig. 3), we use a threshold of Benjamini-Hochberg adjusted $p < 0.01$, corresponding to a more stringent $-\log_{10}(p)$ threshold of 4.8 (up from 4). However, even at a stringent Bonferroni threshold, chromosome 18 is detected and chromosome 12 is quite close to detection.

For the diet-responsive component of the GxEMM scan, we now report the threshold as a "nominal" threshold of $p < 10^{-4}$, and report the BH-adjusted p-value at the chromosome 5 locus (0.126). Methods have been adjusted accordingly.

"Assuming that remaining additive effects explain less phenotypic variance..."

I would move this to the discussion (there is similar material there. Although I would tend to advocate removing it all since it makes a lot of assumptions about the distribution of the un-identified / unknown QTL effect sizes.)

This has been moved to the Discussion and removed from the Results. The language around this estimation has also been changed to reflect that there are likely hundreds or thousands of variants that influence lifespan, albeit with smaller effects than those detected in this manuscript.

"including interaction terms for sex (Shock study) or dietary intervention (DRiDO study)."

The Harrison study also has interaction terms, but is not re-tested?

The only additive effect detected in this study was on chromosome 18 and is almost certainly the same genetic effect re-tested in the DRiDO study. Nonetheless, we re-tested this locus in the Harrison study and did not detect any interaction with dietary intervention. The revised manuscript notes this.

"None of the QTL were found to have interactions with sex or dietary..."

Is this perhaps just a power issue?

Yes, but we can also say that these effects are probably small given the effect sizes of the loci that were detected and the percentage of heritable variation explained by the QTL identified in the study; this is now reflected in the discussion section of the manuscript.

"the detection of QTL unique to individual studies raises the possibility that differences in experimental design..."

Also just statistical power could explain this?

Absolutely; this is now acknowledged in the discussion section.

"different dietary interventions have been shown to have contrasting effects on health-related phenotypes"

Citations?

We've added citations to papers that show contrasting effects of dietary interventions on body weight and lifespan as a function of genetic background.

"This QTL had the largest effect..."

Which QTL is being referred is not 100% clear.

We now refer to this QTL as the "chromosome 5 QTL".

"roughly estimate that at least 12 QTL would be required"

Sure. But it could equally be 1200, right?

Agreed, and this is now addressed in the updated Discussion section.

Figure 1

- Would be useful to describe better in the figure legend the set up for the 4 studies (e.g. that B/C/D are all females).

Figure 1 has been changed so that panels B, C, and D all state that these experiments consist of female mice. The figure legend now explicitly states the experimental design/treatment groups present in each study, including the sex of the animals and the abbreviations for the diet groups.

- Mouse numbers in 1C x-axis legend are wrong (same as 1B)

Great catch! These numbers have been updated in the figure.

- All the abbreviations need to be in the legend (e.g. "2D") so the figure can stand alone.

See the first response to figure 1 - the legend now states the abbreviations for all diet groups for panels B - E.

- Legend states "Vertical bars on each curve indicate censorship events" Not seeing these. Looks like these bars were too small and their opacity too low. We've increased these parameters so that the vertical bars are more visible.

Figure 2

- A-C are not really Manhattan plots; they're QTL profiles.

We now refer to panels A-C as QTL profiles in the figure legends.

- The grayed-out square in the lower right of 2E looks a bit odd.

This has been removed.

- The legend for 2D implies the Chr18 QTL was detected in the Shock study "detected in the Harrison (left), DRiDO (middle), or Shock (left) study"

We've updated the wording in this figure legend to reflect that the chr.18 locus was only detected at genome-wide significance thresholds in the Harrison and DRiDO studies.

Our original goal with the rightmost panel was to see whether the allelic effects of this locus were similar in the Shock study despite it failing to reach genome-wide or nominal significance thresholds.

Figure 3

- A-B are not really Manhattan plots; they're QTL profiles.

We now refer to these plots as QTL profiles.

- Legends to D-I are very repetitive.

The legends are now presented in a compact list format and extraneous information has been removed.

- Particularly in J the tiny horizontal black line looks like part of the "1" CAST allele symbol. I would advocate for a dotted line through the entire of the middle of the plot to indicate where 'additive' would be.

That's a good suggestion; done.

Figure 4

- As mentioned above, it would be useful to define in the legend what "most likely" means statistically.

The legend for Fig. 4A now clarifies which SNPs are highlighted in the variant association mapping plots: "The most likely candidate SNPs, **defined as those with a LOD score within 1LOD of the maximum variant association**, are highlighted in pink."

- Is there anything meaningful to the gene color-coding scheme? Would perhaps help with "seeing" the candidates if those were colored differently than the non-candidates (within QTL but not highlighted by association) and then those not in the QTL (outside the 2-LOD QTL drop).

As it stands, these colors are the default used by the association mapping and plotting functions in R:QTL2 and are meant to help a reader differentiate between the genes within the window, but nothing more. There is concern that using the same colors for genes under vs. next to highlighted SNPs might make it harder to differentiate between genes in the window.

While we make note of specific genes in the main text, we would also like for readers to view these plots fairly agnostically and to bring their own perspective to the lists of genes within the 2LOD intervals of these QTL.

Figure 5

- D needs to be made wider to properly see the bars.

This figure has been adjusted for clarity by increasing the spacing between the data points and increasing the weight of the lines separating the conditions.

- E is not readable at this resolution. Either make much bigger, or delete, or make bigger and put into supplement.

Panel E has been made larger and font sizes have been increased.

"Dietary Restriction (DRiDO):"

Not clear what food type is here; just states "g" but not grams of what.

The methods section for the DRiDO study has been updated to state the type of food the mice received. "G" in the parenthetical descriptions of the dietary interventions now reads "grams" and we have specified that these values reflect the amount of food mice were allotted over the course of the dietary intervention.

"Additive whole-genome scans:"

Not including kinship in the analyses? This would be atypical for DO QTL scans, no?

The text in this methods section has been updated to acknowledge the inclusion of a kinship matrix in all whole-genome scans: **"In each whole-genome scan, kinship was included in the linear mixed effects model as a random effect."**

The methods section titled **"Single-QTL models for diet- and sex-specific loci"** has also been updated to acknowledge the inclusion of kinship in the models used.

"While less conservative, this threshold is more stringent than a previously reported method"

This is a little nebulous. More concrete detail would be useful.

Rather than compare our threshold to the previously reported threshold in Wright, et al. (2022), which was established for a slightly different methodology, we now perform a permutation-based false discovery rate analysis across a range of LOD thresholds for each study, described in the methods section titled *"False discovery rate analysis"*. In the methods section titled "Additive whole-genome scans", we report the FDR corresponding to LOD 6 in each individual study as well as in the meta-analysis.

Given the difference in power among studies, using a consistent FDR threshold arguably makes more sense than using a consistent LOD threshold for the detection of QTL; however, in

practice such an approach has little impact on the results presented in the manuscript. In the methods section titled "Additive whole-genome scans", we note that adjusting FDR thresholds from an average of 0.169 to 0.15 would not impact QTL detection in the individual studies, while adjusting the FDR to 0.10 in the individual studies would only result in a failure to detect two nominally significant QTL on chromosome 7 in the DRiDO study, which are not heavily discussed. The FDR at LOD 6 in the meta-analysis is 0.091; adjusting this FDR to 0.1 would not impact the detection of QTL reported in the manuscript.

March 28, 2025

RE: GENETICS-2025-307990

Dr. Gary A. Churchill
The Jackson Laboratory
Computational Biology
600 Main Street
Bar Harbor, Maine 04609

Dear Gary

We are delighted to inform you that your manuscript titled "Analysis of lifespan across Diversity Outbred mouse studies identifies multiple longevity-associated loci" is acceptable for publication in GENETICS. Many thanks for submitting your research to the journal.

Could you make sure you check Figure 3 please - see the reviewer's comment "On the understanding that Figure 3 is as it was in the original manuscript, but with the edits described in the response to reviews"

The reviewers had a few suggestions for improving the manuscript that you may want to consider. You can view their comments at the bottom of this email.

To Proceed to Production:

1. Format your article according to GENETICS style, as discussed at <https://academic.oup.com/genetics/pages/general-instructions>, and upload your final files at <https://genetics.msubmit.net>.
2. Your manuscript will be published as-is (unedited-as submitted, reviewed, and accepted) at the GENETICS website as an Advanced Access article and deposited into PubMed shortly after receipt of source files and the completed license to publish. Please notify sourcefiles@thegsajournals.org if you do not wish to publish your article via Advanced Access.
3. We invite you to submit an original color figure related to your paper for consideration as cover art. Please email your submission to the editorial office or upload it with your final files. You can submit a small-sized image for evaluation, and if selected, the final image must be a TIFF file 2513px wide by 3263px high (8.375 by 10.875 inches; resolution of 600ppi). Please avoid graphs and small type.

If you have any questions or encounter any problems while uploading your accepted manuscript files, please email the editorial office at sourcefiles@thegsajournals.org.

Sincerely,

Jonathan Flint
Associate Editor
GENETICS

Approved by:
Anthony Long
Senior Editor
GENETICS

note: Please add jnls.author.support@oup.com and genetics.oup@kwglobal.com (or the domains @oup.com and @kwglobal.com) to your email program's "safe senders" list. You will be contacted by both at various points during the production process.

Review comments (if applicable):

Reviewer #2 :

Impressive upgrade. Thanks. This is important work.

Reviewer #3 :

Review of Mullis et al. Revision

Aside from the Figure 3 snafu the revision is greatly improved; I appreciate the effort in responding to the reviews. On the understanding that Figure 3 is as it was in the original manuscript, but with the edits described in the response to reviews, I have no major issues, and only a handful of very minor things for the authors to consider.

Since there are no line numbers the material in quotes below is so the authors can find where my comments come from.

#

"We identified eight loci that contributed significantly..." I would split this sentence into two, ending the 1st sentence after "least one study".

#

"...drive the genetic analyses described herein." Reads a little odd to me. I would revise to "...drive the genetic variation in lifespan."

#

"6% sterilized gain (5K52" You want "grain" here.

#

"...similar to the method described above." Given that what is previously described is that things were bad, maybe add in here that things were not bad in this case?

#

"While less conservative, this threshold..." This section is about FDRs. Would it make things more concise if it was incorporated in the "False discovery rate analysis" section that comes after it?

#

"DRiDO study. t We report" Needs a fix.

#

"corresponding to a 2 LOD drop around each peak position, about each peak marker identified in whole-genome scans." Something is off here.

#

"standard errors95% were computed" Space missing.

#

Anyway this - "the "rqt12" user guide website: https://kbroman.org/qrtl2/assets/vignettes/user_guide.html." - could point to a more permanent site than a personal website? Is it on GitHub maybe?

#

"In the Shock study, genetics explained 15.0% (SE = 15.0%) of variance in lifespan, indicating a large degree of uncertainty." Reads oddly: The SE value is what indicates uncertainty not the h^2 value.

#

"This locus exhibited..." Which locus is being referred to isn't perfectly clear since the previous sentence is about something else.

#

"We therefore expect these loci to deliver" Presumably "these" refers to the 8 mapped loci? Could be clearer.

#

"significance in that dataset.. E, Comparisons" Too many periods.

#

"Figure 3" is actually Figure 5.

#

"study, G, the the second chromosome" Too many "the"s.

#

"DO mice as a function the number of CAST alleles" Missing "of".

#

You map 2 loci to Chr16 and refer to them as the "second chromosome 16 locus" or "16.1" (and so on). I guess I think it would be better to call them something more concrete that doesn't depend on which order one does the analysis. Maybe "2:POSITION_ON_CHR" or something?

#